# DECOUPLING PLANNING AND CONTROL FOR INSTRUCTABLE AGENTS

## ABSTRACT

Recent work on vision-language(-action) agents shows that VLMs are strong at high-level reasoning but struggle to realize plans as reliable low-latency action sequences, while world-model controllers excel at fast observation-to-action control but lack open-ended task guidance. In this work, we combine these strengths by conditioning a learned world-model controller on language so that it can act autonomously at high frequency conditioned on sparse, higher-latency textual instructions generated by vision-language models (VLMs). Our system, Speak-to-Act, includes an instructable controller that autoregressively generates high-frequency actions and can either follow language instructions from an instruction agent, or self-operate in a high-throughput environment. To train controllers to be language-instructable, we relabel segments of controller policy rollouts with instructions and optimize a behavior-cloning objective. Our framework easily supports extension to multi-agent settings that enable agent communication between VLMs using trained controllers as actuators without relying on Multi-Agent Reinforcement Learning algorithms. We report results on various embodied environments and tasks, scaling trends with larger controllers and VLMs, and ablations on instruction cadence, planning frequency, and online vs. offline planning latency. The results show that with our decoupled architecture, Speak-to-Act can flexibly switch to different VLMs and scale well to multi-agents and longer chains of reasoning achieving state-of-the-art performance on six tasks.

## 1 INTRODUCTION

Embodied agents increasingly operate in settings that demand both high-frequency, low-latency continuous control and intermittent, high-level reasoning and planning. For example, consider a long-horizon, multi-player game like Minecraft, where agents can communicate with one another to achieve shared goals. Towards this end, there have been two dominant approaches in building such agents: world-model–based low-level controllers (Hafner et al., 2020; 2023a), often trained with reinforcement learning (RL), and instruction-following large language and vision-language(-action) models (LLMs/VLMs/VLAs) (Zitkovich et al., 2023; Black et al., 2024; Kim et al., 2024; Bjorck et al., 2025). World-model–based RL has shown that compact modeling of latent environment dynamics can support sample-efficient control from pixels (Hafner et al., 2023a). In parallel, instruction-following LLMs/VLMs can synthesize useful decompositions, subgoals, and plans from high-level task descriptions and visual context. Yet, these two capabilities are typically deployed in isolation. VLM-only agents plan well but struggle to realize plans as precise action streams under tight real-time budgets; moreover, as VLMs' output space is limited to text tokens rather than time-critical action streams, evaluating their performance as embodied agents across diverse domains is challenging. Low-level controllers act smoothly and quickly but are difficult to steer with abstract, open-ended instructions.

In this work, we study the problem of grounding language into policy for real-time control coupled with high-level abstraction. We propose Speak-to-Act, a plug-and-play paradigm that decomposes the problem of embodied decision-making into *planning* and *control*. In this paradigm, a pre-trained VLM *planner* maps from environment observations to plans and instructions; instructions are sent to a *controller* which processes a stream of observations and produces and executes low-level actions in real time. The VLM planner plans and issues instructions concurrently with low-level execution, without the need for finetuning. Our core contributions then center around a framework for efficiently training environment-specific controllers that can be paired with any planner to support real-time planning and control. Each

controller is built as a **language-aware world model and policy**, where both representation learning (on the world model) and control (actor and critic heads) are conditioned on a latent representation of an instruction sent by a planner. In contrast with single-task models like Dreamer (Hafner et al., 2020), where a policy is learned indirectly through rewards, our setting allows latent imagination and value learning to incorporate a wide range of abstract goals specified via a natural language instruction. To train controllers to be conditioned on instructions, we perform **post-hoc instruction supervision**, summarizing replay segments into language instructions with a VLM and using them to perform behavior cloning during training. Finally, during inference, we perform **asynchronous instruction**, where the high-level planner runs inference simultaneously as low-level control is executed. This allows a VLM planner to continuously reason and plan in text space while the controller takes actions in the environment.

Our asynchronous instruction framework makes inference scale well in terms of performance and efficiency. Thus, with a learned controller, we can use any number of language planners coupled with shared weights controller copies to scale this framework to several agents. This enables efficient, **multi-agent communication** in real-time and paves the way for human-VLM interaction. We use parameter-shared controllers and let language coordinate at the task level with short broadcast messages while each controller remains a purely reactive, decentralized policy at action time. Our asynchronous instruction mechanism, decoupling instruction cadence from control via a stop/continue handshake, allows planners (VLM) to issue guidance without stalling the control loop. This setup lets us evaluate both (a) how well individual VLMs act as embodied single agents and (b) how well they reason about, coordinate with, and communicate with other agents under real-time constraints.

We evaluate across single- and multi-agent domains and organize the results into two parts:

- Evaluating the trained controller—(a) without instructions, the controller matches standard world-model RL baselines; (b) with instructions; and (c) with instructions from other VLMs that was not used for training.
- Using the controllers to evaluate VLMs' ability to —(a) to act as agents across diverse single-player embodied environments; (b) instruct in multi-agent tasks.

Across these settings, results show that (i) real-time VLM inference significantly improves task success and efficiency versus offline planning; (ii) stronger planners and larger controllers scale well in multi-agent settings, improving performance and efficiency; and (iii) competitive results across benchmarks in both single- and multi-agent regimes achieving state-of-the-art performance on 6 out of 7 tasks.

Our contributions can be summarized as:

- Formulate instructable, real-time control as language-conditioned world-model RL with an autonomous base policy that is override-able by language.
- Propose a post-hoc language annotation pipeline that densifies instruction labels in replay without interrupting data collection and optimization.
- Extend to multi-agent cooperation via shared controllers and language-level coordination.
- Provide a scaling and efficiency study on controller capacity, planner capacity, planning cadence, and wall-clock cost across diverse benchmarks.

## 2 OVERVIEW AND EXPERIMENTAL SETUP

We study the general setting of embodied agents, where an agent, possibly conditioned on a natural language specification of a task, maps observations to low-level actions that influence the state of an environment. We propose to seamlessly adapt existing vision-language models (VLMs) into such embodied language-using agents for arbitrary environments through training of language-conditioned, low-level, and environment-specific control policies.

We operate under an inference setting where embodied agents decompose the problems of reasoning about high-level plans (the role of a *planner* model) and reasoning about low-level execution of those plans (the role of a *controller* model). This decompositional approach to inference explicitly contrasts with existing approaches where adaptation requires fine-tuning of VLMs for the target environment's action space (Driess et al., 2023), or approaches where specialized models are trained to map end-to-end from observations to actions in a target environment including VLA (vision-language-action) models (Zitkovich et al., 2023; Black et al., 2024; Kim et al., 2024; Bjorck et al., 2025) and world models (Hafner et al., 2020; 2023a); and closely follows recent work in language-conditioned robot

control (Ahn et al., 2022), where LLMs are used to map from high-level task specifications to high-level instructions that are then executed by a language-conditioned robot control policy. In our paradigm, the *planner* model takes as input an observation of the environment and maps this to a high-level action, represented as a natural language string. This high-level action specification is then sent to the *controller* model, which maps it and a stream of visual observations to a stream of low-level actions.

This abstraction allows us to plug-in any instruction-tuned VLM as a planner that maps from observations (and an optional high-level task specification) to natural language instructions to be sent to the controller, without requiring any fine-tuning of the model. Because state-of-the-art VLMs are increasingly trained to reason in natural language (Wei et al., 2022) and participate in multi-turn conversations with users (Hurst et al., 2024; Team et al., 2024), our decompositional approach allows us to take advantage of the most capable VLMs in environments that require sophisticated reasoning or communication with other agents. Additionally, decomposing the problems of reasoning about high-level actions and execution of those actions makes it possible to perform these two types of reasoning simultaneously, which significantly reduces agent latency.

**Task.** We consider partially-observable environments specified as $(\mathcal{S}, \mathcal{A}, \Omega, O, P, R, \gamma)$ with a horizon $T$, where $\mathcal{S}$ is the state space, $\mathcal{A}$ is the action space, $\Omega$ is the observation space, $P : \mathcal{S} \times \mathcal{A} \to \Delta^{\mathcal{S}}$ is the transition kernel, $O : \mathcal{S} \to \Delta^{Omega}$ is the observation kernel, $R : \mathcal{S} \times \mathcal{A} \to \mathbb{R}$ is the reward function, and $\gamma$ as the discount factor. Pretrained VLMs struggle when $\mathcal{A}$ is continuously (e.g., high fps games like Minecraft/Atari or Humanoid torques), making direct low level action sequence emission not feasible.

At each environment step, an agent policy $\pi : \Omega \to \Delta^{\mathcal{A}}$ maps from observations to a distribution over actions. We study agents which are compositions of independent *planner* and *controller* agents, respectively $\pi_p : \Omega \to \Delta^{\mathcal{X}}$ and $\pi_c : \Omega \times \mathcal{X} \to \Delta^{\mathcal{A}}$, where $\mathcal{X}$ is the set of all possible natural language utterances. As illustrated in Fig. 1, this planner–controller split is closely related to hierarchical RL and the options framework, where higher levels issue temporally extended commands (Dayan & Hinton, 1992; Sutton et al., 1999; Bacon et al., 2017; Nachum et al., 2018; Levy et al., 2018). We sample actions from $\pi$ given an observation $o$ by first sampling an instruction $x \sim \pi_p(\cdot \mid o)$, then sampling a sequence of low-level actions $\overline{a}$ from $\pi_c$, where $a_i \sim \pi_c(\cdot \mid o_i, x)$. We augment the low-level action space to include an action that indicates the instruction $x$ has been completed; when this is selected, we sample a new instruction from $\pi_p$ given the current observation. When no instruction is issued, we use an empty string to keep the input space consistent. In our experiments, the planner agent $\pi_p$ is any instruction-tuned vision-language model; the controller agent $\pi_c$ is a comparably lighter-weight planner-agnostic recurrent state space model (RSSM) that we train specifically for a target environment.

**Controller model architecture.** We experiment with controller models built on top of the Dreamer recurrent state space model (RSSM) architecture (Hafner et al., 2023a). As the RSSM processes a stream of incoming observations, it maintains a latent state $s_t$, using an action decoder head at each step to generate a distribution over the action space. We modify this base architecture to take as input, in addition to an observation, a representation of an instruction $x$ produced by the planner model. To augment the action space to include instruction completion, we add an additional decoder head that generates a probability that the current instruction has been completed.

**Training.** We alternate between data collection from and optimization of the controller model policy $\pi_c$. During data collection, we sample rollouts without conditioning on an instruction and store them in a FIFO replay buffer. Reward is computed using environment-specific success metrics independent of instruction-following accuracy. To train the controller to be conditioned on instructions, we relabel randomly selected contiguous, variable-length segments of the replay buffer with a language instruction sampled from an existing vision-language model. During optimization, we use the same set of losses as Dreamer (Hafner et al., 2023a), which includes world model reconstruction loss, value loss, actor loss, behavior cloning, and stop token loss. We augment this set with a behavior-cloning loss that maximizes $\pi_c(a_i \mid o_i, x_\tau)$, where $x_\tau$ is the instruction generated for the segment containing $(o_i, a_i)$.

**Evaluation.** We evaluate on tasks ranging from classic single-agent environments well-studied in reinforcement learning (e.g., Atari (Bellemare et al., 2013)) and multi-agent tasks based on real cooperative video games (e.g., Pico Park). Evaluation metrics vary across environments. In addition to measuring total agent success across environments for a variety of planners $\pi_p$, we also evaluate low-level action throughput, and the instruction-following accuracy of $\pi_c$ across different planners. We also measure the influence of planner and controller model size, as well as varying amounts of controller proactivity and synchrony in planner reasoning.

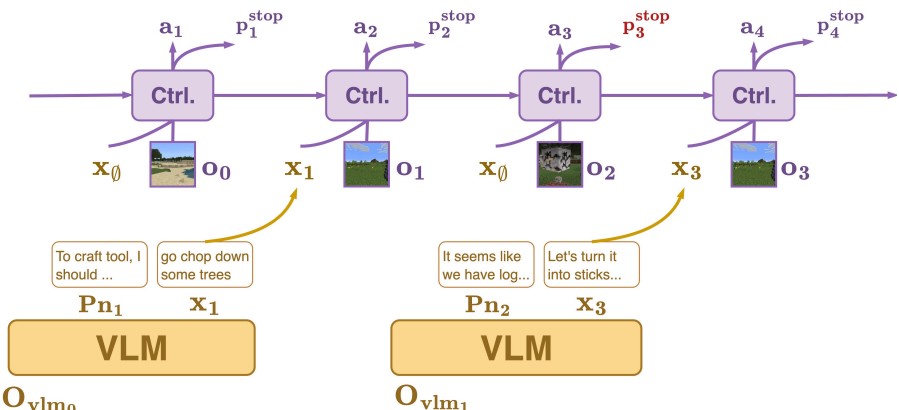

Figure 1: The pipeline that consists of a vision language model and a controller.

## 3 METHOD

### 3.1 DECOUPLING PLANNING AND CONTROL IN INFERENCE

At each step, an agent policy $\pi : \Omega \to \Delta^{\mathcal{A}}$ maps from observations to a distribution over actions. In our framework, $\pi$ is the composition of a planner agent $\pi_p : \Omega \to \Delta^{\mathcal{X}}$ and a controller agent $\pi_c : \Omega \times \mathcal{X} \to \Delta^{\mathcal{A}}$. In practice, we run inference on $\pi_p$ and $\pi_c$ in independent parallel processes and use message passing to update the instruction $x$ on which $\pi_c$ is conditioned, and to notify $\pi_p$ when $\pi_c$ has completed an instruction. This allows us to take advantage of the capabilities of $\pi_p$ beyond high-level decision-making, including reasoning and communication with other agents, while simultaneously executing low-level actions in the environment. It also allows us to easily extend to a multi-agent setting, where we can spawn multiple planners and controller threads corresponding to independent agents that act simultaneously.

**Asynchronous Online Inference.** If we trade off inference between $\pi_p$ and $\pi_c$, such that $\pi_p$ must wait for $\pi_c$ to complete its current instruction before it starts to plan a new one, we waste wall-clock time. Instead, we allow planners $\pi_p$ to run inference asynchronously in the background, given a live observation stream, continuously updating their reasoning. Upon receiving a message that the controller has completed its current instruction, planners are prompted to immediately emit the next instruction from an already-warm draft plan (Fig. 2). Given a draft plan, emitting an instruction requires generating only a few tokens. This simultaneous inference supports both the relatively slow language reasoning of VLMs beneficial to generating optimal instructions, and the fast low-level execution provided by the controller. It also cuts response latency at instruction boundaries, and preserves non-blocking streaming control. Because each planner–controller pair operates largely independently, communication happens online, and planner inference is triggered only by streamed deltas and instruction-completion messages, low-level action throughput scales approximately linearly with the number of pairs until the shared compute or I/O bottleneck saturates.

**Multi-agents Inference.** We extend the framework to cooperative tasks with $n$ agents $\pi^{(i)} = \left( \pi_p^{(i)}, \pi_c \right) ; 1 \leq i \leq n$. Each agent shares controller parameters, but at inference time, these controllers operate on different observation streams, maintain different latent states, and execute different actions. Planners need not be the same type of agent, which allows us to pair, for example, human players with VLM planners. Unlike classic centralized-training/decentralized-execution (CTDE) (Kraemer & Banerjee, 2016), we do not train a separate centralized critic or impose structured role labels. Instead, we let language shape coordination: planners specify high-level goals and coordination happens at this level. The only reward used for learning is the task reward; we do not provide explicit communication rewards or supervision on when to speak. Communication happens as long as the language planners initiate. Agents communicate in discrete rounds aligned with the control step. At each step, the agents can (i) receive inbox message $\text{Inbox}[i]_t$; (ii) instruct the controller to act; (iii) optionally speak. The inbox $\text{Inbox}[i]$ aggregates all peer messages at any given step. In our experiments, all agents share the same system prompt for the planner VLM, which instructs them to process incoming messages and send messages to other agents as if in a dialogue. We observe that with our prompting, agents often use messages to broadcast short updates, like claiming a subgoal, or reporting some blockage on their current subgoal. [1]

---

[1] We give concrete system prompts and detailed schemes in App. A.5.

```
Algorithm 1: Online planning

Inputs: controllers π_c^(1:n), VLMs π_p^(1:n)

Per-agent: Stop[i], Inbox[i] (holds x_t), Plan[i]
(holds plan P_n)

  for i ← 1 to n do
    thread Controller(i):
      while running do
        (a_t^(i), s_t^(i)) ← π_c^(i).step(o_t^(i), x_t^(i)) //
        s_t^(i) ∈ {CONT, STOP}

        if s_t^(i) = STOP then signal(Stop[i]);
        x_{t+1}^(i) ← ∅
        if Inbox[i] has x' then x_{t+1}^(i) ← x'
    thread VLM(i):
      while running do
        if Stop[i] then add x_t^(i) =
        π_p^(i).emit(Plan[i]) to Inbox[i]
        else Plan[i] ← π_p^(i).advance(Plan[i]) //
        background planning

  return Controller streams (o_t^(i) → a_t^(i)); VLM
  maintains P_n in background and emits x_t on STOP.
```

```
Algorithm 2: Offline planning

Inputs: controllers π_c^(1:n), VLMs π_p^(1:n)

Schedule: all pairs advance in lockstep per step
t

  while running do
    for i ← 1 to n do
      observe o_t^(i) // VLM inference every
      control step
      (d_{t+1}^(i), x_t^(i), P_{n,t+1}^(i)) ← π_p^(i).step(d_t^(i),
      o_t^(i), P_{n,t}^(i), Stop[i]) // x_t^(i) ← ∅ if not
      emitted
      // Controller: act given current
      instruction prefix
      (a_t^(i), s_t^(i)) ← π_c^(i).step(o_t^(i), x_t^(i)) //
      s_t^(i) ∈ {CONT, STOP}

      if s_t^(i) = STOP then
        signal(Stop[i])
      t ← t + 1

  return Synchronous loop where VLM and
  controller co-advance every step; no background
  thread, and planning is interleaved with
  control.
```

Figure 2: The algorithm of online planning vs. offline planning.

## 3.2 CONTROLLER ARCHITECTURE

Our controller architecture is based on the recurrent state space model (RSSM) (Hafner et al., 2019a; 2020). This model maintains a latent state during action selection and execution, and uses decoder heads at each step to estimate the next observation, reward, and action to take. Following (Hafner et al., 2023a), we train the observation decoder head with a KL-regularized reconstruction objective and optimize Dreamer-style actor/value heads on imagined rollouts sampled from the latent space. The actor head maps from latent state to action logits,

$$\ell(s_t) \in \mathbb{R}^{|\mathcal{A}|}, \qquad \pi(a \,|\, s_t) = \text{Softmax}\big(\ell(s_t)/\tau\big)_a.$$

where the latent state $s_t$ is determine from previous observations $o_{<t}$ and actions $a_{<t}$.[2] We adapt this base architecture to allow the latent state $s_t$ to be optionally conditioned on an instruction $x_t$ coming from a planner. First, we encode the instruction by tokenizing it and computing

$$e_t = \text{LanguageEncoder}_\ell(x_t) \in \mathbb{R}^{d_\ell},$$

where $x_t = x_\emptyset = \text{empty string}$, if instruction is not present. LanguageEncoder is a pretrained, contrastively learned encoder kept frozen during training. Then the policy becomes

$$\ell(s_t, e_t) \in \mathbb{R}^{|\mathcal{A}|}, \qquad \pi(a \,|\, s_t, e_t) = \text{Softmax}\big(\ell(s_t, e_t)/\tau\big)_a.$$

where we add a projector on top of $e_t$ to concatenate with policy or state input features.

We additionally augment the output space of the controller to include an indicator of whether an instruction has been completed. At each step $t$, the actor emits $p_t^{\text{stop}} = \sigma\big(g(s_t, e_t)\big)$ where $g$ is a small head on top of the latent state and $\sigma$ is the logistic function.

## 3.3 PLANNER

Our framework supports any planner $\pi_p : \Omega \to \Delta^{\mathcal{X}}$ that maps from observations to instructions, where the observation space depends on the environment and can include visual observations and, for multi-task environments, an optional natural language high-level task specification. We experiment with a variety of planners implemented using vision-language models (VLMs) that map from observations to instructions. In addition to mapping from observations to instructions, these VLMs are prompted to, during inference, maintain a memory bank $Mr$ and a partial plan $Pn$. Concretely,

---

[2]Full equations and losses are deferred to App. A.1.

at the VLM's $t'$ inference step, it takes as input a previous state $(Sp, Mr_{t'-1}, Pn_{t'-1}, O_{<t'}, p_{t'-1}^{\text{stop}})$, where $Sp$ is the system prompt sent to the VLM, $O_{<t'}$ is the sequence of previous observations, and $p_{t'-1}^{\text{stop}}$ is the most recent probability that the controller $\pi_c$ has assigned to instruction completion. It maps these inputs to a new state of the memory bank $Mr_{t'}$ and a new partial plan $Pn_{t'}$, and, if $p_{t'-1}^{\text{stop}}$ is true, an instruction $x$ to be sent to the controller $\pi_c$. Crucially, as discussed in Sec. 3.1 the VLM inference timesteps $t'$ are not necessarily synchronous with the controller's timesteps $t$, which supports online VLM planning for a subsequent instruction while the agent is executing its previous instructions in the environment. In offline variant $t'$ will be the same as $t$ ignoring $p_{t'-1}^{\text{stop}}$).

In multi-agent settings, except for the per-agent reasoning and instruction, we add communications between agents in two modes: (1) centralized and (2) decentralized. Given a set of VLMs $[n] = i \in \{1,...,n\}$, for each VLM inference step, any VLM with index $i$ chooses to or not to initiate at most one message to VLM with index $j$, noted as message $m_{ij}$. In centralized mode, the message sending agent is fixed, resulting in agent $h$ sending messages to other agents $i \in [n] \setminus h$, noted as message $m_{hi}$.

## 3.4 TRAINING

We train the controller parameters by alternating between collecting rollouts conditioned on the current policy, and optimizing the policy using both reward, world model, and instruction-following behavior-cloning objectives.[3]

Our set of losses can thus be considered a combination of two types of objectives: (a) to have the controller follow instructions accurately, and (b) to have the controller's policy assign high probability to observations that the planner will be exposed to at inference time. To achieve (a), we use behavior cloning objectives that train the controller policy to assign high probability to action sequences when conditioned on instructions that describe those sequences. We acquire data for behavior cloning by sampling action sequences on-policy from the controller and labeling them with an instruction by using a vision-language model. We thus must jointly optimize objective (b) to align instruction-labeled action sequences with those the controller samples and executes at inference time.

**Behavior cloning with post-hoc instruction annotation.** During learning, we use a FIFO replay buffer (Mnih et al., 2015) $\mathcal{D}$ with capacity $|\mathcal{D}| = 1024$ that stores tuples

$$\tau_t \equiv (o_t, a_t, r_t, e_t, \text{complete}_t, \text{done}_t)$$

where $o_t$ is the observation used for action selection, $a_t$ is the action sampled from the policy, $r_t$ is the reward this action received from the environment, $e_t$ is an (optional) embedding of an instruction, $\text{complete}_t$ is a label indicating whether the action $a_t$ completed the (optional) instruction, and $\text{done}_t$ is a binary variable indicating whether the episode has terminated at time $t$.[4]

To acquire instruction embeddings $e_t$ to store in the replay buffer, we annotate segments of the buffer tuples online as the controller generates each action. First, a segment length $L$ is randomized within an integer interval. Then, we sample a start index $t \sim \text{Uniform}\{1,...,|\mathcal{D}|-L\}$ and extract the segment $(o_{t:t+L-1}, a_{t:t+L-1}, r_{t:t+L-1}, c_{t:t+L-1})$. For a sampled buffer of length $|\mathcal{D}|$, we sample a set of non-overlapping intervals $\mathcal{I} = \{[u_k, v_k]\}_{k=1}^K$. Each interval $[u_k, v_k]$ comprises the start and endpoint of a segment to annotate. For each such interval, we use a vision-language model (VLM) to summarize the action sequence, including the observations $o_{u_k:v_k}$, into a high-level instruction $x$. Concretely, the VLM summarizer takes in task-specific prompts (details in App. A.5), equally spaced frames, and packed action sequence text that belong to the interval. We then encode $x$ into a vector $e$ as mentioned in Sec. 3.2, and pair all tuples $t \in [u_k : v_k]$ with $e$. For $t = v_k$, we set $\text{complete}_t$ to 1, otherwise we set the value to 0.

During training, we treat $a_{u_k:v_k}$ as a label for the instruction $x$ conditioned on observations $o_{u_k:v_k}$, and optimize a behavior cloning loss over labeled intervals:

$$\mathcal{L}_{\text{BC}} = -\lambda_{\text{BC}} \sum_{[u_k,v_k] \in \mathcal{I}} \sum_{t=u_k}^{v_k} \log \pi (a_t \,|\, s_t, e_t).$$

---

[3]We use Dreamer-style control losses for optimizing the RSSM via reward and its world model estimation as defined in App. A.1.

[4]When collecting a rollout of length $T$, different timesteps $t$ may be associated with different instruction embeddings $e_t$ and instruction completion indicators $\text{complete}_t$, whereas $\text{done}_t$ is true if and only if $t = T$.

To learn to mark instructions as complete, we also apply a binary cross-entropy loss on $p_t^{stop}$ using the label $\text{complete}_t$: $\mathcal{L}_{\text{stop}} = -\sum_t \Big(\text{complete}_t \log\big(p_t^{\text{stop}}\big) + (1-\text{complete}_t)\log\big(1-p_t^{\text{stop}}\big)\Big)$.

During training, not entire buffer is annotated, we sample non-overlapping intervals that sum up to 50% of the buffer. This allows the controller to autonomously act according to a policy that optimizes the reward provided by the environment, $R$. Moreover, the annotation does not cost large training efficiency overhead as it reuses logged trajectories and runs in parallel with optimization. When no instruction is provided, we set $e_t = \mathbf{0}$. We (optionally) support a strict instruction-required mode that architecturally guarantees no actions are executed without a valid, non-completed instruction.

**Final Training Objective.** The overall objective augments the base objective with behavior cloning:

$$\mathcal{L} = \mathcal{L}_{\text{model}} + \lambda_V \mathcal{L}_{\text{value}} + \lambda_A \mathcal{L}_{\text{actor}} + \mathcal{L}_{\text{BC}} + \mathcal{L}_{\text{stop}}$$

where $\mathcal{L}_{\text{model}}$, $\mathcal{L}_{\text{value}}$, and $\mathcal{L}_{\text{actor}}$ are defined by the base RSSM training algorithm (Hafner et al., 2023a).

## 4 EXPERIMENTS

### 4.1 ARCHITECTURE AND TRAINING

For the controller, we instantiate a language-conditioned DreamerV3-style (Hafner et al., 2023b) agent.[5] For our main results, we use a controller with 800M parameters. We use MiniLM-L6-H384-uncased (Wang et al., 2020) as the language encoder. Environment rollouts are sampled conditioned on null language embeddings during collection ($e_t = \mathbf{0}$). For post-hoc instruction annotation (Section 3.4), we sample random segments of randomized length between 1 to 20 steps from the buffer. Sampled subintervals receive tokenized instructions and masks, while gaps carry empty instruction and no behavior cloning loss.

### 4.2 EVALUATION

Each evaluation consists of 100 episodes (or the standard evaluation protocol of the particular task). During evaluation, we do not use exploration noise ($\varepsilon = 0$) and use temperature $\tau = 0.8$ for stochastic policies. Atari (Kaiser et al., 2019) uses sticky-actions and no-op starts; MC Diamond follows the MineRL (Guss et al., 2019) ObtainDiamond task with evaluation inventory checks; Crafter (Hafner, 2021) uses the official success metrics; DMLab (Beattie et al., 2016) uses the levels from the `explore_goal_locations` suite. For Overcooked, we use the `Asymmetric Advantages` and `Cramped Room` layouts with scripted partner policies for single-agent tests. PicoPark single-agent levels measure puzzle completion. Overcooked multi-agent uses cross-play (Carroll et al., 2019). PicoPark evaluates cooperative puzzle solving with synchronized actions and shared success metrics.[6] MindCraft (White et al., 2025) evaluates task success on cooking/construction/crafting.[7]

### 4.3 VLMS AS PLANNERS

Table 1 shows the main results, comparing different VLMs as planners with several baseline agents. We train one controller per environment that runs efficiently as training does not require VLM reasoning. In the bottom rows, we include the performance of environment-specific baseline agents as reported by prior work. For multi-agent tasks, we also train a multi-agent policy following Kuba et al. (2021). In the middle rows, we report two baseline systems: using GPT-4o end-to-end as an agent, taking as input observations and directly generating low-level actions; and a controller-only baseline, where we run inference with the controller without conditioning on instructions, which provides a lower-bound of agent performance. Finally, we experiment with four popular vision-language models as planners.

Our approach to decompose decision-making into *planning* and *control* consistently outperforms prior environment-specific approaches on nearly all environments with all tested VLM planners. Nearly all tested VLM planners also significantly outperform an end-to-end GPT-4o agent in all environments. Using GPT-4o to generate instructions executed by a trained controller, rather than having GPT-4o

---

[5]See hyperparameters and other details in App. A.3. Complete architecture scales are reported in App. A.4.

[6]Pico Park (`https://picoparkgame.com/en/`) is a 2-8 player 2d cooperative platformer puzzle game.

[7]Detailed tasks descriptions are in App.B

Table 1: Main results. Each cell reports the results using the task's official metrics where available. For Pico Park, we report success rate over 8 levels across all sampled episodes.

| | | Single-Agent Tasks | | | | Multi-agent Tasks | | |
|---|---|---|---|---|---|---|---|---|
| | | **Atari** | **Diamond** | **Crafter** | **DMLab** | **Overcooked** | **PicoPark** | **Mindcraft** |
| **VLMs** | Gemma-3-27B (Team et al., 2025) | 880 | 9.7 | 13.8 | 71 | 187.4 | 68.9 | 53.0 |
| | llava-v1.6-34b (Liu et al., 2023a) | 862 | 9.9 | 12.8 | 74 | 187.2 | 63.5 | 51.6 |
| | Qwen-VL-2.5-72B (Team, 2024) | 878 | 11.1 | 13.4 | 77 | 192.3 | 68.4 | 58.5 |
| | GPT-4o (Hurst et al., 2024) | 891 | 11.7 | 14.1 | 76 | 193.2 | 70.1 | 70.2 |
| **Ablation** | GPT-4o (w/o controller) | 670 | 10.4 | 8.7 | 56 | 180.4 | 50.3 | 50.2 |
| | Controller-Only | 809 | 8.2 | 12.6 | 67 | 170.2 | 30.7 | 40.0 |
| **Prev. SOTA** | Dreamerv3 (Hafner et al., 2023a) | 811 | 8.6 | 10.5 | 65 | - | - | - |
| | Voyager (Wang et al., 2023) | - | 11.8 | - | - | - | - | - |
| | MARL Baseline (Kuba et al., 2021) | - | - | - | - | 182.5 | 50.8 | 44.9 |
| | Mindcraft (Claude) (White et al., 2025) | - | - | - | - | - | - | 49.0 |

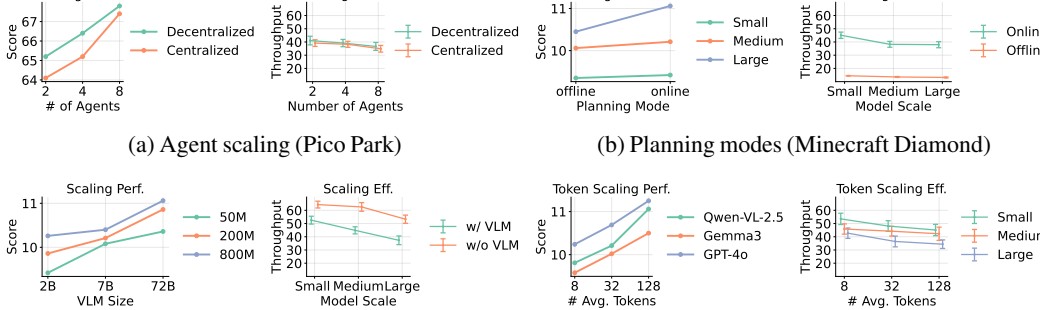

(a) Agent scaling (Pico Park)  (b) Planning modes (Minecraft Diamond)

(c) Model size scaling (Minecraft Diamond)  (d) Reasoning scaling (Minecraft Diamond)

Figure 3: Each subfigure shows a pair of results: *Performance* (left) and *Efficiency* (right). Models scales for controller and VLMs presented are: small (50M, 2B), medium (200M, 7B), and large (800M, 72B).

directly generate low-level actions itself, significantly improves its performance in all tasks. This demonstrates the efficacy of our decompositional approach: instead of expecting a VLM to be able to generate low-level actions directly, we allow it to instead reason about high-level plans, and leave the control problem to the controller. Finally, providing a controller instructions generated by a VLM planner significantly improves agent performance in all environments for nearly all planners. Our results allow us to compare different VLMs as agents in new games. For example, we find that larger models (GPT-4o, Qwen-VL-2.5-72B) outperform the smaller VLMs tested.

### 4.4 SCALABILITY AND EFFICIENCY

We evaluate the framework's performance and efficiency (throughput as environment steps per second) under (1) multi-agent settings with varying communication modes and agent numbers, (2) online vs. offline VLM planning, (3) scaling model size, and (4) reasoning density by average number of reasoning tokens produced during online VLM planning. We experiment with three model scales using Qwen-VL-2.5 (Team, 2024): (1) 50M controller with 2B VLM; (2) 200M controller with 7B VLM; (3) 800M controller with 72B VLM.

**Multi-agents scaling:** Fig. 3 (a, left) shows that performance improves consistently on Pico Park, which relies heavily on collaboration, as we increase the number of agents, and that decentralized control surpasses centralized control. Fig. 3 (a, right) shows that adding more agents to the environment does not significantly affect inference efficiency.

**Planning mode:** Fig. 3 (b, right) shows that online planning, where a VLM planner can compose partial plans while the controller is executing a previous instruction, significantly improves agent inference efficiency, while Fig. 3 (b, left) shows incremental performance increase when the planner is able to compose partial plans given real-time observations.

**Controller and VLM size scaling:** We assess scaling by sweeping controller parameter counts (50M/200M/800M) and VLM backbones (2B/7B/72B). Evaluation follows the achievement score criterion within a 10k step in-game budget. Fig. 3 (c, left) shows that larger VLMs and controllers generally improve success, and do not significantly harm throughput Fig. 3 (c, right).

| Instruction Mode | VLM Planner | | |
|---|---|---|---|
| | Gemma | Qwen | GPT |
| Fixed Cadence | 8.86 | 9.32 | 9.60 |
| Fully controlled by VLM | 9.89 | 10.12 | 10.52 |
| Proactiveness decided by VLM | 10.12 | 10.23 | 10.75 |

Table 2: Single-agent tests under different instruction modes. (Minecraft Diamond)

| Annotator | Eval Planner | | |
|---|---|---|---|
| | Gemma | Qwen | GPT |
| Gemma | 9.6 | 9.2 | 10.2 |
| Qwen | 9.7 | 9.9 | 9.9 |
| GPT | 9.8 | 9.6 | 10.0 |

Table 3: Cross-Planner train × eval matrix (Minecraft Diamond).

**Reasoning density:** We experiment with limiting the amount of planning tokens the VLM can include in its partial plan to $\{8,32,128\}$ tokens per instruction it issues. Fig. 3 (d, left) reports achievement scores (%) under each setting for three VLMs. We also show that the efficiency is not significantly harmed when increasing the number of reasoning tokens Fig. 3 (d, right).

### 4.5 GENERALIZATION

**Instruction modes (Minecraft):** We contrast four instruction regimes (Table 2): (1) *Fixed cadence*: planners issue instructions every $K{=}16$ steps; (2) *VLM-decided proactiveness* (the VLM emits instructions only when its uncertainty exceeds a threshold, with a minimum gap of 32 steps); (3) *Fully controlled by VLM*. We report achievement scores for three planners.

**Cross-planner generalization:** We evaluate how training under one planner transfers to another at test time by forming a train × eval planner matrix (Table 3). Each cell reports the achievement score when trained with instructions annotated by the row model and evaluated with the column planner.

## 5 RELATED WORK

Our controller and training setup builds on top of recurrent state-space models (RSSMs) and imagination-based control, leveraging latent word representations to plan and execute tasks. PlaNet introduced a stochastic–deterministic latent transition for planning from pixels (Hafner et al., 2019b). Dreamer and its successors learn behaviors by backpropagating value gradients through imagined trajectories in the latent space (Hafner et al., 2020; 2023a). Our multi-agent setting uses parameter sharing at test time, replicating a single set of weights across agents while keeping per-agent recurrent states, which is similar to established MARL works (Terry et al., 2021; Chu & Ye, 2017).

Instruction-following vision–language models (VLMs) such as LLaVA (Liu et al., 2023b), Gemma (Gemma Team et al., 2024), Qwen-VL and Qwen2-VL (Bai et al., 2023; Yang et al., 2024) provide a practical interface for specifying high-level intent. We treat the language encoder as an auxiliary input to the world model and policy, similar to recent VLMs that couple visual encoders with LLMs for multimodal instruction following. Similar to us, recent work uses high-level agents like LLM to combine with low-level robotics policy, but makes assumptions about the sharing of the knowledge of LLM and the controller (Ahn et al., 2022). Open-ended agents such as Voyager show skill libraries and LLM-driven exploration in Minecraft (Wang et al., 2023). Our RL-Instruction-Mixed approach complements these by grounding instruction in a learned world model and a low-latency streaming policy that can act with or without language.

Previous works on LLM-driven tasks rely on games like Minecraft, which tolerate short horizons and high-latency environments. We also evaluate on diverse challenging tasks: Atari/ALE for pixel-control, DeepMind Lab for 3D first-person tasks, Crafter for open-world survival, and MineRL's ObtainDiamond are all high-throughput environments that challenge LLM-based methods. Multi-agent environments PettingZoo, Melting Pot, and Overcooked-AI all require strong visual and high-level reasoning that present out-of-distribution challenges to world models that don't have a high-level brain.

## 6 CONCLUSION

We have introduced Speak-to-Act, a framework that takes advantage of both the low latency of RL controllers and the high-level reasoning ability of VLMs. Our framework is competitive with state-of-the-art agentic frameworks in solving complex tasks while maintaining low latency. Moreover, we have shown its flexibility and effectiveness in extending to a multi-agent setting without the need for a meticulously designed multi-agent algorithm.

# 7 REPRODUCIBILITY STATEMENT

We provide a complete recipe to re-create our results. The overall online inference paradigm is included in Sec. 3.1. The controller architecture is included in Sec. 3.2 and the hyperparameter and scale configs are described in App. A.3 and App. A.4. The planner algorithm is discussed in Sec. 3.3 and the detailed prompting is included in App. A.5. The training process is discussed in Sec. 3.4 and its details are expanded in App. A.3. Evaluation is first discussed in Sec. 4.2 and later expanded along with tasks details in App. B. Most datasets and models (except for proprietary VLMs) are open source. We will open-source our code upon publication.

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

## A MODEL DETAILS

### A.1 WORLD-MODEL AND CONTROL DETAILS

**RSSM transition and inference.** Conditioning on language, the deterministic transition and stochastic prior/posterior are

$$h_t = f_\theta(h_{t-1}, z_{t-1}, a_{t-1}, e_{t-1}),$$

$$p_\theta(z_t \,|\, h_t) = \mathcal{N}\big(\mu_\theta(h_t), \mathrm{diag}\,\sigma_\theta^2(h_t)\big),$$

$$q_\phi(z_t \,|\, h_t, o_t, e_t) = \mathcal{N}\big(\mu_\phi(h_t, o_t, e_t), \mathrm{diag}\,\sigma_\phi^2(h_t, o_t, e_t)\big).$$

We denote the model state as $s_t = (h_t, z_t)$ with $z_t \sim q_\phi(\cdot)$ during training and $z_t \sim p_\theta(\cdot)$ during imagination.

**Decoders and predictors.** From $s_t$ we decode/predict:

$$
\begin{aligned}
& p_\theta(o_t \,|\, s_t) && \text{(observation decoder)}, \\
& p_\theta(r_t \,|\, s_t) && \text{(reward model)}, \\
& p_\theta(c_t \,|\, s_t) && \text{(continuation/discount model with } c_t \in (0,1)), \\
& p_\theta(x_{t+1} \,|\, s_t) && \text{(next-instruction predictor; optional teacher forcing on } x_t).
\end{aligned}
$$

The instruction head is trained only when $m_{t+1} = 1$; otherwise its loss is masked out.

**Model learning objective.** Over a sequence $t = 1{:}T$, the model loss combines reconstruction/prediction terms with a KL regularizer:

$$\mathcal{L}_{\mathrm{model}} = \sum_{t=1}^{T} \mathbb{E}_{q_\phi}\Big[ -\log p_\theta(o_t \,|\, s_t) - \lambda_r \log p_\theta(r_t \,|\, s_t) - \lambda_c \log p_\theta(c_t \,|\, s_t) - \lambda_x m_{t+1} \log p_\theta(x_{t+1} \,|\, s_t)\Big]$$

$$+ \beta \sum_{t=1}^{T} \mathrm{KL}\big(q_\phi(z_t \,|\, h_t, o_t, e_t) \,\big\|\, p_\theta(z_t \,|\, h_t)\big),$$

with weights $\lambda_r, \lambda_c, \lambda_x \geq 0$ and KL scale $\beta$.

### A.2 DREAMER-STYLE CONTROL IN LATENT SPACE

Given a learned world model, we learn a policy and value on latent states conditioned on language:

$$\pi_c(a_t \,|\, s_t, e_t), \qquad V_\psi(s_t, e_t).$$

**Imagined rollouts.** Starting from posterior states $s_t$ on real trajectories, we 'imagine' $H$-step futures using the prior dynamics and current policy:

$$\tilde{s}_{t+1} \sim p_\theta(\cdot \,|\, \tilde{h}_{t+1}), \tilde{h}_{t+1} = f_\theta(\tilde{h}_t, \tilde{z}_t, \tilde{a}_t, \tilde{e}_t), \tilde{a}_t \sim \pi_\eta(\cdot \,|\, \tilde{s}_t, \tilde{e}_t), \tilde{r}_t \sim p_\theta(\cdot \,|\, \tilde{s}_t), \tilde{c}_t \sim p_\theta(\cdot \,|\, \tilde{s}_t),$$

where $\tilde{e}_t$ is the instruction embedding available to the agent during imagination (e.g., last known instruction embedding, or an imagined instruction from $p_\theta(x_{t+1} \,|\, s_t)$).

**Value targets via $\lambda$-returns with continuation.** Define the continuation as the learned discount $\gamma \tilde{c}_\tau \in [0,1]$. The $\lambda$-return is

$$\hat{G}_\tau^\lambda = \tilde{r}_\tau + \gamma \tilde{c}_\tau\Big((1-\lambda)V_\psi(\tilde{s}_{\tau+1}, \tilde{e}_{\tau+1}) + \lambda \hat{G}_{\tau+1}^\lambda\Big), \quad \hat{G}_{t+H}^\lambda = V_\psi(\tilde{s}_{t+H}, \tilde{e}_{t+H}).$$

**Actor and value losses.** We optimize the value to regress to the return and the actor to maximize it through imagined trajectories:

$$\mathcal{L}_{\mathrm{value}} = \sum_{\tau=t}^{t+H-1} \big\| V_\psi(\tilde{s}_\tau, \tilde{e}_\tau) - \mathrm{stopgrad}(\hat{G}_\tau^\lambda) \big\|_2^2,$$

$$\mathcal{L}_{\mathrm{actor}} = -\sum_{\tau=t}^{t+H-1} \mathbb{E}_{\tilde{a}_\tau \sim \pi_\eta}\Big[\mathrm{stopgrad}(\hat{G}_\tau^\lambda)\Big] - \alpha \sum_{\tau=t}^{t+H-1} \mathcal{H}(\pi_\eta(\cdot \,|\, \tilde{s}_\tau, \tilde{e}_\tau)),$$

with entropy scale $\alpha \geq 0$. Gradients backpropagate through the imagined dynamics (world model) as in Dreamer.

Table 4: Model-scale configurations.

| Scale (tag) | rssm.deter | rssm.hidden | rssm.classes | depth | units |
|---|---|---|---|---|---|
| 50M (size50m) | 4096 | 512 | 32 | 32 | 512 |
| 200M (size200m) | 8192 | 1024 | 64 | 64 | 1024 |
| 800M (size800m) | 24576 | 3072 | 192 | 192 | 3072 |

### A.3 MODEL HYPERPARAMETERS

We use MiniLM-L6-H384-uncased (Wang et al., 2020) as the language encoder. For small controller as an example, we instantiate a language-conditioned DreamerV3-style (Hafner et al., 2023b) agent with a CNN encoder (4 convolutional blocks; channels [32,64,128,256], stride 2, SILU), an RSSM with a deterministic GRU core of size $512$ and a diagonal-Gaussian stochastic latent $z_t \in \mathbb{R}^{32}$, and decoders for observation/reward/continuation as MLPs (two layers, $512$ units, SILU). The actor and value heads operate in latent space and use three-layer MLPs with $512$ units (SILU). The policy head produces action logits for discrete control or mean/scale for continuous control, depending on the environment action space. We follow dreamer to use KL balancing with scale $\beta{=}1.0$ and free-nats $1.5$, reconstruction/prediction weights $(\lambda_r,\lambda_c,\lambda_x){=}(1,1,1)$, imagined horizon $H{=}15$ (as above), and continuation-based $\lambda$-returns. The language pathway pools a frozen text encoder into an embedding $e_t$ that conditions both the world model and control; when absent, a learned null embedding is used. Optimizer and update schedule follow the hyperparameters above (Adam, $3\mathrm{e}{-}4$, clip $40$, 1:1 world-model/control updates).

### A.4 CONTROLLER SCALES

Table. 4 show controller scales from 50M, 200M, to 800M. These configs are adapted from Dreamerv3 (Hafner et al., 2023a) and re-used the 50M and 200M configuration.

### A.5 PROMPT DETAILS

In this section, we list the prompts we used for training and inference including multi-agents. The summarizer prompt for GPT-4o as shown in Fig. 4 is used for post-hoc training language annotation in Sec. 3.4. The inference prompt as shown in Fig. 4 is used for online reasoning and issuing instructions as mentioned in Sec. 3.1. The multi-agent prompt as shown in Fig. 4 is used for prompting multi-agents communication. The full prompt is simply appending this to the single-agent inference prompt. Note that the math notations are all replaced by actual inputs or in-context examples.

## B TASKS DETAILS

We list and describe the evaluation tasks in details here totaling 7 environments. Every environment can be discussed from its goal, the observation space, evaluation protocol, and the metric.

**Arcade Learning Environment (ALE)**

- **Goal:** Achieve high scores across Atari 2600 titles using standard ALE evaluation.
- **Obs/Act:** Raw visual frames with common preprocessing (frame skip/stack); full primitive action set.
- **Protocol:** Follow the evaluation practices popularized in recent Atari work, e.g., Kaiser et al. (2019), including fixed evaluation episodes and capped frames per episode; when using sticky actions or ALE "game flavours," also follow the ALE protocol paper.
- **Metric:** Game score per episode; report means (and variance) over seeds.

**MineRL ObtainDiamond**

- **Goal:** Obtain a diamond in survival Minecraft starting from scratch.

You are a concise action summarizer for training labels. Read inputs IN THIS ORDER:

1. TaskGuidance G (what succeeds in this environment) and a short Message M (any extra note),
2. sampled visual Frames F[t : t+L-1] with timestamps,
3. the executed Actions A[t : t+L-1] with timestamps.

Goal

produce ONE short imperative instruction x that, if given before t, would make a competent controller reproduce A[t : t+L-1]. Keep it atomic and task-grounded.

Rules

- Focus on intent, not low-level joystick/button spam.
- Refer only to what is visible/achieved in Frames and consistent with Actions.
- No hallucinations; if ambiguous, pick the MINIMAL instruction that explains A.
- 6–18 tokens; avoid conjunction chains; one clause; present tense verb first word.
- If the segment ends at completion, align x to that subgoal's completion boundary.

Output JSON ONLY

```
{
  "instruction": "<single-line imperative>",
  "justification": "<8-20 token rationale tied to frames/actions>",
  "segment": {"t_start": t, "t_end": t+L-1}
}
```

[USER]

TaskGuidance (G): {G}
Message (M): {M}
Frames F[t:t+L-1]: {FRAME_LIST_OR_CAPTIONS}
Actions A[t:t+L-1]: {ACTION_LIST}

Figure 4: Training summarization prompt.

- **Obs/Act:** $64 \times 64$ first-person RGB plus discrete inventory observations; actions combine continuous camera control with discrete navigation, mining/crafting/smelting interactions.

- **Protocol:** Episode terminates on death, diamond obtained, or 18,000-frame (15 min) limit. Competition evaluation averages performance over 500 episodes on fixed but unseen seeds; strict train-from-scratch compliance.

- **Metric:** Shaped milestone reward (e.g., logs, planks, pickaxes, iron, diamond), summed per episode; tie-break by fewest episodes to last milestone.

**Crafter**

- **Goal:** Survive and progress in a procedurally generated 2D open world by unlocking semantically meaningful achievements (e.g., find water, craft tools, defeat enemies).

- **Obs/Act:** Local top-down pixel observations showing surroundings and inventory status; discrete actions for movement, interaction, crafting, sleeping, placing.

- **Protocol:** Two tracks: with extrinsic rewards and reward-free. Agents get a fixed budget of 1M environment steps (commonly also reported at 5M in baselines). Success rates computed across the entire training run to emphasize sample efficiency.

- **Metric:** *Crafter score* is the geometric mean of the 22 achievement success rates, emphasizing breadth/depth of capabilities.

**DeepMind Lab: `explore_goal_locations`**

```
[SYSTEM = Sp]
You are the planner module π_p running ASYNCHRONOUSLY with a low-latency controller
π_c.

Maintain
 • Reasoning memory M_r (brief scratch notes).
 • Partial plan P_n (bullet list of next subgoals → micro-steps).

You receive a live observation stream O_{<t'} and a stop signal probability p_stop_{t'-1} from
π_c.

Emit a NEW instruction x ONLY when either (a) p_stop_{t'-1} is true/high (≥ 0.5), or (b) you
proactively decide the current plan needs revision. Otherwise ADVANCE your background
plan silently.

Constraints
 • Instructions are 6–18 tokens, single-clause, imperative, specific and verifiable from
   observations.
 • Never block control; if unsure, issue the safest non-destructive next step.
 • Keep M_r and P_n ultra-compact; prefer deltas over full rewrites.

Output JSON ONLY

{
 "emit": <true|false>,
 "instruction": "<if emit=true, the instruction; else empty>",
 "reasoning_update": "<<=40 tokens delta for M_r>",
 "plan_update": ["<<=12 tokens step 1>", "<step 2>", "..."],
 "confidence": <0.0-1.0>
}

[USER]

Previous state

- M_r(t'-1): {M_r_prev}
- P_n(t'-1): {P_n_prev}
- p_stop_{t'-1}: {p_stop_prev}

New context

- O_{<t'} (latest frames/summaries): {OBS_STREAM_SNIPPET}
- Task spec (if any): {TASK_SPEC}
```

Figure 5: Single-agent prompt.

- **Goal:** First-person 3D navigation in maze-like levels to discover and reach goal locations under partial observability.

- **Obs/Act:** RGB first-person frames; continuous look and movement controls.

- **Protocol:** Standard DM-Lab evaluation with fixed episode caps; commonly evaluated on held-out maps (unseen layouts), sometimes with environment-provided debug info *only* for visualization/analysis (not agent inputs).

- **Metric:** Episode return (goals found/reached), success rate, and path efficiency; report means over seeds/maps.

**Overcooked-AI**

- **Goal:** Two-player cooperative cooking (deliver soups quickly) with strong *zero-shot coordination* (ZSC) to unseen partners and layouts.

- **Obs/Act:** Gridworld state or egocentric features; discrete actions (move, interact, pick/place).

```
[SYSTEM = Sp_multi_agent_addon]
You are planner i in a cooperative team of n agents. You may send at MOST ONE short
message per step.

Inbox processing
first read Inbox[i] (messages addressed to you or broadcast). Update M_r and P_n accordingly.

Communication modes
 • Decentralized: you decide when to message.
 • Centralized (hub=h): only agent h broadcasts role/assignment messages; others reply
   sparsely.

Messaging rules
 • Keep messages ≤ 25 tokens.
 • Prefer structured intents: {claim:<subgoal>}, {status:<brief>}, {block:<issue>}, {request:
   <help on X>}, {handoff:<asset/role>}.
 • Addressing: use @all for broadcast or @j for a specific agent j.
 • Do not restate observations verbatim; send actionable deltas that change teammates' plans.

Augmented OUTPUT (append to (2)'s JSON)

"comm": {
 "send": <true|false>,
 "to": "@all" | "@j" | "@hub",
 "message": "<one line following the intent tags above>"
}
```

Figure 6: Multi-agent prompt (to be appended to single-agent prompt).

- **Protocol:** Evaluate on canonical layouts (*Cramped Room*, *Asymmetric Advantages*, *Coordination Ring*, etc.) and out-of-distribution layouts; cross-play with behavior-cloned human models and held-out agents; human studies where applicable.
- **Metric:** Team return (deliveries/time), success rate, and ZSC scores (cross-play averages across unseen partners/layouts).

**Pico Park (Co-op Puzzles)**

- **Goal:** Complete short, cooperative puzzle-platforming levels that demand synchronized actions (e.g., stacking, shared switches, tethered movement).
- **Obs/Act:** Platformer state with simple discrete controls for 2–8 players (levels scale with player count).
- **Protocol:** Curate a fixed subset of multiplayer levels (e.g., 48 classic levels) and evaluate multiple seeds/player configurations; require that all agents reach the goal to clear a level.
- **Metric:** Level completion rate and median completion time across the suite; optionally, coordination error counts (drops, desyncs).

**MindCraft (Minecraft Multi-Agent Collaboration)**

- **Goal:** Multi-agent embodied collaboration on *Cooking*, *Crafting*, and *Construction* tasks via language-enabled coordination.
- **Obs/Act:** First-person Minecraft control with inventories; agents exchange natural-language messages; tasks provide recipes/blueprints and split resources/knowledge across teammates.
- **Protocol:** Procedurally generated tasks per category; for construction, initialize agents with blueprints and disjoint materials/skills; for cooking/crafting, vary recipe complexity and information asymmetry ("Hell's Kitchen" variants).

- **Metric:** Average success rate for Cooking & Crafting; Construction uses an edit-distance alignment between the built structure and the target blueprint; overall score averages category scores.

