# OpenReview forum: "Decoupling Planning and Control for Instructable Agents"
_ICLR.cc/2026/Conference — Submitted to ICLR 2026_

### Official Review · Reviewer_Kt4V · 2025-10-27

**Soundness:** 3
**Presentation:** 3
**Contribution:** 3
**Rating:** 6
**Confidence:** 4

**Summary:**

This paper introduces the Speak-to-Act framework, which effectively decouples high-level planning (VLM) from low-level control (SRRM) to construct steerable embodied agents. Its core innovation lies in training environment-specific controllers through post-hoc VLM-generated instruction labels, enabling asynchronous real-time execution and effortless scalability to multi-agent settings, i.e., without relying on multi-agent reinforcement learning (MARL). Experimental evaluations cover both single-agent and multi-agent tasks, achieving state-of-the-art (SOTA) performance on 6 out of 7 benchmarks, and demonstrating promising scalability trends for larger controllers and planners.

**Strengths:**

1.	Ingenious asynchronous online inference mechanism: It effectively prevents slow VLM reasoning from blocking the control loop, i.e., the planner can prewarm a draft in the background and only outputs a concise suffix once the instruction completion signal is triggered. This design enables real-time multi-agent scalability, with near-linear growth in action throughput, without requiring VLM fine-tuning or MARL integration.
2.	Efficient and extensible post-hoc instruction labeling: By summarizing replay segments with a VLM, the framework significantly increases label density without interfering with data collection. Combined with hybrid optimization of the Dreamer objective and behavior cloning loss, it ensures that the controller retains the autonomy of its base policy while allowing flexible instruction overrides.
3.	Highly modular architecture: The framework allows seamless substitution of different VLMs, supports hybrid use of human and VLM planners in multi-agent environments, and generalizes across diverse domains (e.g., from Atari to Pico Park).

**Weaknesses:**

1.	Limited originality of core components: Although the overall integration is creative, elements such as language-conditioned RSSM and VLM-based planning are not novel; the asynchronous inference and post-hoc labeling techniques, while practical, primarily represent engineering optimizations rather than conceptual breakthroughs.
2.	Fragile instruction pipeline due to VLM dependency: Both training summarization and inference planning heavily depend on the VLM, which may amplify its inherent errors, e.g., noisy summary instructions could introduce bias during behavior cloning.
3.	Weak empirical support for SOTA claims: The paper claims SOTA results across multiple tasks but provides only high-level metrics (e.g., success rate, throughput, instruction accuracy) without detailed baseline comparisons. While multi-agent results show scalability of the planner, the work lacks deeper comparison with centralized-training/decentralized-execution MARL methods (e.g., QMIX) or end-to-end VLA models.
4.	Overly idealized scalability assumptions: claimed linear scalability depends on shared controllers and parameters. However, in asymmetric multi-agent environments, individual fine-tuning may be required for each agent.
5.	Incomplete efficiency analysis: Although the framework relies on background VLM drafts and shared controllers, it does not quantify the actual computational overhead (e.g., VLM token generation cost in multi-agent settings) or potential I/O bottlenecks.

**Questions:**

1.	Beyond parity with the Dreamer baseline, how does Speak-to-Act outperform end-to-end VLA methods (e.g., RT-2) in terms of latency and success rate?
2.	experiments suggest that real-time VLM outperforms offline planning, but by what margin? Are there quantitative results correlating controller size (e.g., parameter count) with performance?
3.	How does the system handle conflicts between VLM-generated instructions and environment rewards? In sparse-reward settings, how well does it perform when dense success indicators are unavailable?

---

> ### Author Response · Authors · 2025-11-15
>
> Thanks for your review! When you mention “The paper claims SOTA results across multiple tasks but provides only high-level metrics…”, could you please clarify what other metrics you believe we should report? Please clarify soon so we can prioritize computing such metrics so we can address your concern.

---

> ### Author Response · Authors · 2025-11-27
> **Response to Reviewer Kt4V (1/4)**
>
> **W.1 Novelty of Speak-to-Act vs prior language-conditioned RSSM and VLM planning**
>
> We fully agree that hierarchical / planner–controller structures have an existing body of work, and we build on that line of work. Our contribution is not the abstract idea of “a planner plus a controller”, but a specific, end-to-end framework that supports plug-and-play evaluation of any VLM-based planners in new environments, including multi-agent environments, via domain-specific, instruction-conditioned world-model controllers.
> - The controller is trained purely on environment reward and world-model losses, augmented with post-hoc instruction behavior cloning that uses on-policy rollouts annotated after the fact by a VLM. At inference time, the planner can be any VLM (Gemma, Qwen, GPT-4o, etc.), without any domain-specific or output-space fine-tuning. In contrast, existing hierarchical structures are often trained end-to-end on specific domains, including the finetuning of the high-level planner, while we emphasize the ability to swap out different pretrained planners to evaluate them independently
> - Prior VLA and CTDE works such as DEPS, LS-Imagine, and JARVIS-1 combine language and control into embodied agents that take actions to achieve high-level goals, but these approaches typically (i) couple planning and control more tightly (e.g., they require a full VLM forward pass for every single low-level action generated, while in our case we only require a forward pass of the much smaller low-level controller to generate each low-level action), (ii) train a monolithic VLA while our training process only replies on a smaller controller model, or (iii) focus on single-agent or just specific to multi-agents. Our contribution is a framework that supports deployment and evaluation of arbitrary VLM planners in combination with low-level world-model controllers to result in embodied agents, which, in comparison with a VLA approach:
>   - requires no planner fine-tuning,
>   - can be shared across multiple agents, and
>   - achieves strong empirical performance and efficiency across domains.
>
> **W.2 Robustness to noisy VLM instructions and dependence on VLM quality**
>
> We agree that more explicit ablations would clarify the role of instruction quality. Our current results already hint at robustness to annotation source (Table 3 in original paper shows similar performance when training with Gemma/Qwen/GPT annotations and evaluating with different planners). Concretely, controllers trained with Gemma-, Qwen-, or GPT-based annotations yield very similar absolute returns and, more importantly, induce nearly the same ordering over planners; the only deviation is that Qwen-annotated controllers slightly favor Qwen planners, while GPT-annotated controllers produce rankings closest to those from Gemma annotations.

---

> ### Author Response · Authors · 2025-11-27
> **Response to Reviewer Kt4V (2/4)**
>
> **W.3 Empirical support for SOTA claims vs MARL and VLA baselines**
>
> **Table 1. Benchmarking with VLA/MARL Baselines.**
> Method                | Atari | Diamond | Crafter | DMLab | Overcooked | Pico Park | MindCraft
> --------                                    | ------| --------| --------| ------| ---------- | ----------| ----------
> Gemma-3-27B            | 880   | 9.7     | 13.8    | 71    | 187.4      | 68.9      | 53.0
> LLaVA-v1.6-34B         | 862   | 9.9     | 12.8    | 74    | 187.2      | 63.5      | 51.6
> Qwen-VL-2.5-72B       | 878   | 11.1    | 13.4    | 77    | 192.3      | 68.4      | 58.5
> GPT-4o           | 891   | 11.7    | 14.1    | 76    | 193.2      | 70.1      | 70.2
> **Ablations**                           |
> Qwen-VL-2.5-72B (Direct VLM Finetune)   | 581   | 10.1    | 7.6    | 45    | 150.1      | 30.6      | 38.2
> GPT-4o (w/o controller)         | 670   | 10.4    | 8.7     | 56    | 180.4      | 50.3      | 50.2
> Controller-Only                             | 809   | 8.2     | 12.6    | 67    | 170.2      | 30.7      | 40.0
> DreamerV3    | 811   | 8.6     | 10.5    | 65    | -          | -         | -
> MARL Baseline       | -     | -       | -       | -     | 182.5      | 50.8      | 44.9
> **Zero-shot or Promptable**
> MindCraft (Claude)    | -     | -       | -       | -     | -          | -         | 49.0
> Voyage  | -     | 11.8    | -       | -     | -          | -         | -
> RT-2 [1]  (public reproduction)            | 457   | 6.2     | 4.7     | 36    | 124.7  |  34.0  | 33.1
> DEPS [3]               | - | 9.4    |  -  | -  |    -  | -   | 45.6
> LS-Imagine [4]           | -  | 9.6    | -   | -  | -    | -  | 50.1
> JARVIS-1 [6]           | -  | 12.3    | -   | -  | -    | -  | 54.1
> **Finetuned from same Rollout**                           |
> RT-2 [1]  (public reproduction)           | 685   | 9.2     | 8.5     | 58    | 157.5  |  44.2  | 48.3
> DEPS [3]              | - | 10.6    |  -  | -  |    -  | -   | 56.8
> LS-Imagine [4]           | -  | 10.9    | -   | -  | -    | -  | 59.3
> QMIX [5] (public reproduction)              | -   | -    | -    | -  | 187.2      | 58.5      | 53.2
>
> We show more baselines and recent VLA in Table 1 on VLA and MARL baselines. Following the suggestions, we included comparisons with the following VLA and CTDE methods: RT-2, DEPS, LS-Imagine, JARVIS-1, and QMIX. We find that they are competitive in their original domains but our framework outperforms them in general, especially in the more realistic multi-agent collaborative domains, while JARVIS-1 outperforms us in Minecraft Diamond as it is a meticulously designed framework for Minecraft only. (Note: We included RT-2 and QMIX’s open reproduction as the baselines. We won’t be able to include results using RT-H as it's not open-source. We can’t run DEPS and LS-Imagine on all tasks as their preset prompts are exclusive for Minecraft.)
>
> **W.4 Scalability assumptions and extension to asymmetric multi-agent roles**
>
> Our current tasks are indeed role-symmetric, and we use parameter sharing for controllers across agents.
> We do not claim to have solved arbitrary asymmetric role settings.
> However, our framework is naturally extensible to asymmetric multi-agent environments in two ways:
> - Role-specific prompts for VLM planners. Different agents can be given distinct role descriptions (e.g., “builder”, “scout”) in the planner prompt, while still sharing the same controller backbone.
> - Role conditioning for controllers. Should asymmetries be strong enough that sharing a single controller becomes limiting, one can add a small role embedding input to the controller while largely reusing the same architecture and training pipeline.
> We will adjust the scalability discussion to clarify that our empirical claims are made for symmetric settings and that asymmetric roles are a natural extension rather than a proven property.
>
> **Q.1 Latency and success rate compared to end-to-end VLA (e.g., RT-2)**
>
> **Table 2. Inference Speed with VLA Baselines**
>
> Method                      | Throughput Env steps/s
> --------------------------- | ---------------
> RT-2 [1]   | 16.2
> DEPS [3]      | 20.5
> LS-Imagine [4]        | 23.7
> JAVIS-1  | 26.0
> Speak-to-Act (Qwen-2.5-VL)       | 31.7
>
> Our framework improves both success and practical latency compared to end-to-end VLA models such as RT-2. In terms of performance, Table 1 shows that pairing GPT-4o with our controller consistently outperforms a finetuned RT-2 reproduction on all shared tasks. In terms of inference cost, our framework runs the controller at every environment step, while invoking the VLM only every K steps to update partial plans. This avoids a full VLM forward pass at 10–60 Hz and keeps controller-only throughput close to a Dreamer-V3 baseline. Note that we replace all VLMs used in these frameworks with Qwen-2.5-VL for a fair comparison.

---

> ### Author Response · Authors · 2025-11-27
> **Response to Reviewer Kt4V (3/4)**
>
> **Q.2 Quantitative margins for online vs offline planning and controller/VLM size scaling**
>
> Figure 3 (b) already compares offline/synchronous vs online/asynchronous planning:
> - Asynchronous planning significantly boosts environment throughput (right panel), because the controller continues acting while the planner reasons in the background.
> - Performance also improves on large scale variants by 0.6, because online planning allows the VLM to revise instructions based on more up-to-date observations.
>
> Figure 3 (c) complements this by studying controller and VLM size scaling:
> - Larger controllers (50M,200M, to 800M) and larger VLMs (2B,7B, to 72B) generally improve success rates (left panel) without significantly reducing throughput (right panel), suggesting that our asynchronous design keeps inference efficient even at larger scales.
> We will make this connection explicit in the text and add a short paragraph summarizing the observed margins.
>
> **Q.3 Handling conflicts between VLM instructions and environment rewards, especially in sparse-reward settings**
>
> To clarify, the environment reward is always the ground-truth high-level task reward, and we never introduce a language-based reward.
> - In dense-reward settings, if the VLM annotator issues suboptimal instructions, the RL objective pushes the controller toward reward-maximizing behavior, even if this involves occasionally training on instructions paired with inaccurate demonstrations.
> - In sparse-reward settings, the controller still leverages Dreamer-style world-model imagination and value gradients; instructions provide an auxiliary label that helps organize behavior into interpretable segments, but do not change where reward is located.

---

> ### Author Response · Authors · 2025-11-27
> **Response to Reviewer Kt4V (4/4)**
>
> **References**
>
> [1] Zitkovich, Brianna, et al. "Rt-2: Vision-language-action models transfer web knowledge to robotic control." Conference on Robot Learning. PMLR, 2023.
>
> [2] Belkhale, Suneel, et al. "RT-H: Action Hierarchies using Language." Robotics: Science and Systems. 2024.
>
> [3] Wang, Zihao, et al. "Describe, explain, plan and select: interactive planning with llms enables open-world multi-task agents." Advances in Neural Information Processing Systems 36 (2023): 34153-34189.
>
> [4] Li, Jiajian, et al. "Open-World Reinforcement Learning over Long Short-Term Imagination." The Thirteenth International Conference on Learning Representations.
>
> [5] Rashid, Tabish, et al. "Monotonic value function factorisation for deep multi-agent reinforcement learning." Journal of Machine Learning Research 21.178 (2020): 1-51.
>
> [6] Wang, Zihao, et al. "Jarvis-1: Open-world multi-task agents with memory-augmented multimodal language models." IEEE Transactions on Pattern Analysis and Machine Intelligence (2024).
>
> We sincerely thank the reviewers for the feedbacks and will do additional experiments or clarification upon request.

---

### Official Review · Reviewer_eR51 · 2025-10-31

**Soundness:** 2
**Presentation:** 2
**Contribution:** 2
**Rating:** 2
**Confidence:** 4

**Summary:**

This work introduces Speak-to-Act, a system that combines the high-level reasoning of Vision-Language Models (VLMs) with the fast, low-level control of world-model controllers. The system allows a VLM to provide sparse, high-latency language instructions to a high-frequency controller, which is trained via behavior cloning to execute these commands. This decoupled architecture achieves state-of-the-art performance on six tasks, scales well, and easily extends to multi-agent scenarios without complex reinforcement learning.

**Strengths:**

Strength:

1: The overall exposition and logical flow of the paper are clear and well-structured.

2: The paper significantly outperforms the baselines and demonstrates impressive single-task performance.

3: The proposed architecture is technically reasonable and well motivated. Its design shows promising scalability with respect to both model capacity (number of parameters) and multi-agent deployment, suggesting potential for broader application at larger scales.

4: The strong cross-planner generalization further supports the soundness and robustness of the proposed framework.

**Weaknesses:**

1: The proposed planner–controller architecture, which is presented as a core contribution of this paper, is not sufficiently novel. Similar hierarchical designs have been explored in several prior works, such as DEPS ,LS-Imagine, JARVIS-1.

2: The paper lacks comparison and discussion with recent VLA(vision-language-action)-related works, such as RT-H (and other vision-language-action models). Including these would provide a clearer picture of how the proposed framework relates to existing VLA systems.

3: The number of effective baselines reported in the main results is too limited, making it difficult to assess the relative performance of the proposed method.

4: The paper lacks an ablation study on the controller architecture. It would be informative to explore alternative designs, such as a Transformer-based Controller, to verify whether the observed performance gains stem from the proposed structure itself or from general modeling capacity.

5: The paper lacks discussion on multi-task generalization.

6: The multi-agent experiments lack detailed analysis, such as case studies or a comparative discussion of the centralized vs. decentralized communication modes and their respective advantages.

**Questions:**

See the weakness.

---

> ### Author Response · Authors · 2025-11-15
>
> Thanks for your review! When you mention “The number of effective baselines reported in the main results is too limited, making it difficult to assess the relative performance of the proposed method.”, which specific baselines are you looking for? We are planning to add more baselines including VLA models. Please clarify if there are additional baselines besides these we should run, and we will prioritize evaluating them.
> Additionally, when you mention “The paper lacks discussion on multi-task generalization.”, could you be more specific about what you mean by “task” here? Please clarify soon so we can determine if addressing this requires us to perform additional experiments or analysis. Thank you!

---

> ### Author Response · Authors · 2025-11-27
> **Response to Reviewer eR51 (1/3)**
>
> Thank you for your careful review. We will address these points below.
>
> **W.1 Novelty relative to DEPS, LS-Imagine, JARVIS-1 and other planner–controller architectures**
>
> We fully agree that hierarchical / planner–controller structures have an existing body of work, and we build on that line of work. Our contribution is not the abstract idea of “a planner plus a controller”, but a specific, end-to-end framework that supports plug-and-play evaluation of any VLM-based planners in new environments, including multi-agent environments, via domain-specific, instruction-conditioned world-model controllers.
> - The controller is trained purely on environment reward and world-model losses, augmented with post-hoc instruction behavior cloning that uses on-policy rollouts annotated after the fact by a VLM. At inference time, the planner can be any VLM (Gemma, Qwen, GPT-4o, etc.), without any domain-specific or output-space fine-tuning. In contrast, existing hierarchical structures are often trained end-to-end on specific domains, including the finetuning of the high-level planner, while we emphasize the ability to swap out different pretrained planners to evaluate them independently
> - Prior VLA and CTDE works such as DEPS, LS-Imagine, and JARVIS-1 combine language and control into embodied agents that take actions to achieve high-level goals, but these approaches typically (i) couple planning and control more tightly (e.g., they require a full VLM forward pass for every single low-level action generated, while in our case we only require a forward pass of the much smaller low-level controller to generate each low-level action), (ii) train a monolithic VLA while our training process only replies on a smaller controller model, or (iii) focus on single-agent or just specific to multi-agents. Our contribution is a framework that supports deployment and evaluation of arbitrary VLM planners in combination with low-level world-model controllers to result in embodied agents, which, in comparison with a VLA approach:
>   - requires no planner fine-tuning,
>   - can be shared across multiple agents, and
>   - achieves strong empirical performance and efficiency across domains.
>
> **W.2/W.3 Comparisons to recent VLA systems (RT-2, RT-H, and related) and more baselines**
>
> **Table 1. Benchmarking with VLA/MARL Baselines.**
> Method                | Atari | Diamond | Crafter | DMLab | Overcooked | Pico Park | MindCraft
> --------                                    | ------| --------| --------| ------| ---------- | ----------| ----------
> Gemma-3-27B            | 880   | 9.7     | 13.8    | 71    | 187.4      | 68.9      | 53.0
> LLaVA-v1.6-34B         | 862   | 9.9     | 12.8    | 74    | 187.2      | 63.5      | 51.6
> Qwen-VL-2.5-72B       | 878   | 11.1    | 13.4    | 77    | 192.3      | 68.4      | 58.5
> GPT-4o           | 891   | 11.7    | 14.1    | 76    | 193.2      | 70.1      | 70.2
> **Ablations**                           |
> Qwen-VL-2.5-72B (Direct VLM Finetune)   | 581   | 10.1    | 7.6    | 45    | 150.1      | 30.6      | 38.2
> GPT-4o (w/o controller)         | 670   | 10.4    | 8.7     | 56    | 180.4      | 50.3      | 50.2
> Controller-Only                             | 809   | 8.2     | 12.6    | 67    | 170.2      | 30.7      | 40.0
> DreamerV3    | 811   | 8.6     | 10.5    | 65    | -          | -         | -
> MARL Baseline       | -     | -       | -       | -     | 182.5      | 50.8      | 44.9
> **Zero-shot or Promptable**
> MindCraft (Claude)    | -     | -       | -       | -     | -          | -         | 49.0
> Voyage  | -     | 11.8    | -       | -     | -          | -         | -
> RT-2 [1]  (public reproduction)            | 457   | 6.2     | 4.7     | 36    | 124.7  |  34.0  | 33.1
> DEPS [3]               | - | 9.4    |  -  | -  |    -  | -   | 45.6
> LS-Imagine [4]           | -  | 9.6    | -   | -  | -    | -  | 50.1
> JARVIS-1 [6]           | -  | 12.3    | -   | -  | -    | -  | 54.1
> **Finetuned from same Rollout**                           |
> RT-2 [1]  (public reproduction)           | 685   | 9.2     | 8.5     | 58    | 157.5  |  44.2  | 48.3
> DEPS [3]              | - | 10.6    |  -  | -  |    -  | -   | 56.8
> LS-Imagine [4]           | -  | 10.9    | -   | -  | -    | -  | 59.3
> QMIX [5] (public reproduction)              | -   | -    | -    | -  | 187.2      | 58.5      | 53.2
>
> We show more baselines and recent VLA in Table 1 on VLA and MARL baselines. Following the suggestions, we included comparisons with more VLA and CTDE methods. We find that some are competitive in their original domains but our framework outperforms them in general. JARVIS-1 outperforms us in Minecraft Diamond as it is a meticulously designed framework for Minecraft only. (Note: We included RT-2 and QMIX’s open reproduction as the baselines. We won’t be able to include results using RT-H as it's not open-source. We can’t run DEPS and LS-Imagine on all tasks as their preset prompts are exclusive for Minecraft.)

---

> ### Author Response · Authors · 2025-11-27
> **Response to Reviewer eR51 (2/3)**
>
> **W.4 Ablation on controller architectures (Transformer, RNN, value-only)**
>
> **Table 2 Ablation on controller architectures.**
> Controller architecture    | Minecraft Diamond
> ---------- | -----
> Dreamer-V3-style RSSM (ours)  | 11.0
> Transformer-based world-model controller [7]     | 11.2
> RNN policy (model-free)     | 9.7
> Value-only world model       | 10.2
>
> While our current paper instantiates the controller as Dreamer-V3, our framework is architecture-agnostic. We do clarify that our framework is architecture-agnostic: in principle, one could plug in a Transformer-based controller or other model-based RL backbones while keeping the same planner interface and training procedure. We have experiments with alternative controllers such as TransDreamer [7] and RNN policies; in Table 2 these results suggest similar trends, and we will include these ablations in the revision.
>
> (Note: The transformer-based world-model controller [7] is a sequence model over latent state with similar training and imagination. It performs on-par with controllers used in our framework but is more computationally heavy. The RNN policy (model-free) is an RNN policy without explicit world model, a.k.a, no imagination rollouts. Moreover, the value-only world model is a world model used only for value estimation with a separate policy network. )
>
> **W.5 Multi-task generalization**
>
> Our current experiments evaluate single-task training across multiple environments (Atari, Crafter, DMLab, Minecraft Diamond, Overcooked, Pico Park, MindCraft), and In Pico Park experiments. We train on levels 1-4 and test on 5-8. In this game, different levels have significantly different mechanics. For example, level 2 requires players to push down a wall, level 5 requires players to find buttons everywhere to unlock stages. This, along with our other results, do shows that our framework can generalize to unseen mechanics.
> We do not claim multi-task training in the sense of “train one controller jointly on many environments and test zero-shot on unseen and significantly different games.” We will clarify this in the paper.
>
> **W.6 Case studies and centralized vs. decentralized multi-agent communication**
>
> Section 4.4 currently reports:
> Scaling trends in multi-agents experiments in the Figure 3 (a) plots as we vary agent count and communication mode, showing that decentralized communication yields higher success while throughput remains essentially stable as agents are added.
> We agree that more qualitative analysis would make the results clearer. Here is our findings:
> - In the centralized mode, successful episodes are characterized by a clear structure: the planner agent issues short messages (“Subgoal: mine gold”, “Now farm an apple”) while the worker agent mostly returns status updates (“iron 6, gold 8 acquired; returning to surface”). Failures typically arise when the hub becomes a bottleneck (e.g., delayed subgoals), leaving workers idle or doing redundant exploration.
> - In the decentralized mode, successful episodes rely on early role negotiation. Agents usually take roles (“I’m on wood and apples”, “I’ll handle the cave”) and send status (“furnace ready at spawn”, “ores in chest”). When episodes fail, we usually see either missing or conflicting claims, e.g., two agents implicitly chasing the same subgoal.
>
> We do not have space to include the examples in the rebuttal. However, we will revise the paper appendix for concrete examples.

---

> ### Author Response · Authors · 2025-11-27
> **Response to Reviewer eR51 (3/3)**
>
> **References**
>
> [1] Zitkovich, Brianna, et al. "Rt-2: Vision-language-action models transfer web knowledge to robotic control." Conference on Robot Learning. PMLR, 2023.
>
> [2] Belkhale, Suneel, et al. "RT-H: Action Hierarchies using Language." Robotics: Science and Systems. 2024.
>
> [3] Wang, Zihao, et al. "Describe, explain, plan and select: interactive planning with llms enables open-world multi-task agents." Advances in Neural Information Processing Systems 36 (2023): 34153-34189.
>
> [4] Li, Jiajian, et al. "Open-World Reinforcement Learning over Long Short-Term Imagination." The Thirteenth International Conference on Learning Representations.
>
> [5] Rashid, Tabish, et al. "Monotonic value function factorisation for deep multi-agent reinforcement learning." Journal of Machine Learning Research 21.178 (2020): 1-51.
>
> [6] Wang, Zihao, et al. "Jarvis-1: Open-world multi-task agents with memory-augmented multimodal language models." IEEE Transactions on Pattern Analysis and Machine Intelligence (2024).
>
> [7] Chen, Chang, et al. "Transdreamer: Reinforcement learning with transformer world models." arXiv preprint arXiv:2202.09481 (2022).
>
> We sincerely thank the reviewers for the feedbacks and will do additional experiments or clarification upon request.

---

### Official Review · Reviewer_jmVV · 2025-11-01

**Soundness:** 2
**Presentation:** 2
**Contribution:** 3
**Rating:** 4
**Confidence:** 4

**Summary:**

The paper proposes Speak-To-Act, an agent framework for action generation given vision observations and optionally language instructions. The proposed method decouples the planner from controller, using a VLM to serve as a planner, and a Recurrent State Space Model (RSSM) as the controller for action predictions. The paper highlights that since the interface between the planner and the controller is natural language (embeddings), the VLMs could be switched and plugged in without fine-tuning. The paper also highlighted that the planner and the controller are asynchronous, which allows high throughout action outputs possible. The proposed method could also be extended to the Muti-agent setting through optionally natural language output, paving the way for future human-AI collaborations. The proposed method is based on DREAMER, following a similar sets of training loss, including world model reconstruction loss, value loss, actor loss, behavior cloning loss, and stop token loss. The paper conducted experiments in both the single-agent setting and the multi-agent setting, spanning across 7 environments, and multiple baseline models. The paper also evaluated the scalability of the method, low level action throughout, as well as the instruction-following accuracy of the planners. The paper also conducted a series ablation studies and generalizability of the method.

**Strengths:**

- The paper proposed a method Speak-to-Act, that decouples agent planning from action generation
- The proposed method is asynchronous, which supports high-frequency action throughput
- The proposed method leverages natural language as the interface, which makes it easier to plot in and switch different VLM planners
- The paper conducted numerous experiments across multiple environment, planner models, against multiple baselines, and demonstrated improved performance and throughput

**Weaknesses:**

1. There are a lot of great content in the work. Some of the highlights including decoupling and asynchronous planner + controller, plug and play VLMs without fine-tuning, evaluating different VLMs, controller with optional instruction following, single player VS multi players, and much more... It would be helpful to focus on just a subset of the main contributions and structure the experiments and narration around them. At the moment, the experiment results touch a little bit of each point without a much needed in depth analysis and a coherent story. For example, is the goal to evaluate different VLMs? to enable high throughput asynchronous action outputs? To study multi-agent collaborations?

2. The proposed method seems to be an extension of the DREAMER work to introduce optional language instructions with very similar settings, training paradigms, model architectures, etc.

**Questions:**

1. Figure 3: if one of the goals is to highlight the higher action throughput of Speak-to-Act, would it be a stronger piece of supporting evidence if the comparison happens between synchronous VS asynchronous models? (compared to centralized VS decentralized, planning modes, etc)
2. Table 1: is missing quite a bit of performance report from prev.SOTA. Would be helpful for comparison if the models were rerun on the same tasks under similar settings
3. Table 2: how to do the results from the table support the section title "Generalization"

---

> ### Author Response · Authors · 2025-11-27
> **Response to Reviewer jmVV (1/4)**
>
> Thank you for your detailed and constructive review. We will address your concerns as below.
>
> **W.1 Novelty relative to DEPS, LS-Imagine, JARVIS-1 and other planner–controller architectures**
>
> Our work is motivated by the fact that it is difficult to evaluate and deploy instruction-tuned vision-language models (VLMs) as agents completing grounded, embodied tasks, for example, single-player games like Minecraft or Atari, or in multi-player games like Pico Park or Overcooked, without domain-specific finetuning. It’s difficult to directly deploy VLMs in these settings, because success in these environments requires (a) processing a continuous stream of observations as input, at a rate of many images per second; and (b) generating a long sequence of low-level actions, where the action space differs widely across environments, and is certainly not in the pre-training or instruction-tuning data that the VLM was trained on. (Note: although our primary motivation is to support the benchmarking of existing pretrained VLMs on embodied tasks rather than training a domain-specific model that achieves optimal performance on a specific task, even if we consider finetuning VLMs for specific domains, action in such environments significantly stresses their context limits as sequences become long quickly with multiple outputs per timestep; additionally, requiring end-to-end inference of an entire large VLM at each timestamp increases latency; our results also show that this amount of compute is not required to reach top performance if we decouple planning and action.)
>
> The inability to directly deploy VLMs as agents limits our abilities to fully understand and evaluate core competencies of such VLMs that they have ideally acquired through pretraining and instruction tuning: their ability to reason about high-level problems grounded in an embodied environment and, in multi-agent tasks, the ability to coordinate with other agents through the use of language. **Our core contribution is to directly address this limitation by decoupling the problems of planning and control. We supply VLMs with language-based interfaces to the environment in the form of small domain-specific, VLM-agnostic controller models.** Here, the VLM being tested is not expected to take in streaming observation inputs, or generate low-level actions. Instead, it periodically observes new environment states and sends instructions to a low-level controller. In multi-agent environments, it also sends language messages to other agents to coordinate their actions.

---

> ### Author Response · Authors · 2025-11-27
> **Response to Reviewer jmVV (2/4)**
>
> Continuation of **W.1 Novelty relative to DEPS, LS-Imagine, JARVIS-1 and other planner–controller architectures**
>
> Each of our experimental results shows the efficacy and feasibility of our approach, as well as the use of our setup to evaluate several existing popular VLMs on a number of embodied tasks. Most of our development was performed on Minecraft Diamond (as this also offers the most other comparisons with existing approaches, such as Dreamerv3, Voyager, LS-Imagine, and DEPS), so most of our experimental results are focused on this domain.
>
> - **Existing VLMs struggle to be directly deployed as agents in embodied environments.** We verify that the strongest VLM we tested, GPT-4o, struggles to perform well in most of our target tasks, when expected to directly interface with the environment. (Table 1, row GPT-4o (w/o controller)). In contrast, GPT-4o performs significantly better on all environments when paired with a controller as its interface to the environment (Table 1, row GPT-4o). Similarly, the low performance on Mindcraft by the Claude-based agent from White et al. (2025). These results suggest that we are underestimating the abilities of existing VLMs to serve as embodied agents when we expect them to perform high-level reasoning, communication, and additionally continuous observation and out-of-distribution low-level control.
>
> - Several of our experimental results verify the design of our controller architecture and training setup:
>
>   - Table 1 verifies that our controller, trained with a Dreamerv3-style objective augmented with post-hoc instruction-following via demonstrations, achieves performance similar to base Dreamerv3 without the instruction-following objective (see lines Controller-Only and Dreamerv3). Thus, this objective does not hurt the controller’s ability to map from observations to actions.
>   - Table 3 shows that the choice of VLM used for post-hoc instruction annotation during controller training does not significantly influence conclusions about relative ranking of VLMs paired with that controller (except for Qwen, which results in a controller that boosts the performance of a Qwen planner; in our main experiments, we use a controller trained with GPT-4o annotations, whose produced VLM ranking agrees with the controller trained with Gemma annotations).
>   - Our additional results on instruction-following accuracy verify that trained controllers accurately follow over 90% of instructions issued by a VLM planner. This means the controllers serve as an effective interface for VLMs that cannot generate low-level actions at the rate or granularity expected by the environment.
>
> - Our remaining experiments use our proposed benchmarking/evaluation setup to examine the performance of existing VLMs as planners in single-agent and multi-agent embodied environments.
>   - Table 1 includes results on seven tasks for four popular VLMs (Gemma, LLava, Qwen-VL, and GPT-4o). In general, GPT-4o outperforms the other models in all environments, followed by Qwen-VL, Gemma, and Llava. However, GPT-4o still has a significant margin to achieve in its tasks; most easy to interpret is the results from Pico Park and Mindcraft, where multi-agent teams of GPT-4o planners achieve success in only 70% of episodes.
>   - We use our setup to evaluate several variations in how VLM planners interact with the controller.
>     - In the multi-agent Pico Park task, we evaluate how performance scales with the number of agents, and whether one VLM is used to send instructions to multiple controllers (centralized), or whether multiple VLMs are each supplied one controller and must communicate with each other to coordinate their actions (decentralized). We find that decentralized control results in significantly higher success rates. See Figure 3a.
>     - We evaluate how VLM size influences both performance and efficiency on Minecraft Diamond. Figure 3(b) compares offline (synchronous) vs. online (asynchronous) planning with the same planner/controller pair and shows that asynchronous planning yields a consistent gain in score while substantially increasing environment throughput, since the controller continues acting while the planner reasons in the background. Figure 3(c) then studies model-size scaling: as we increase controller size (50M → 200M → 800M) and VLM size (2B → 7B → 72B), success rates monotonically improve while throughput remains essentially flat, indicating that our asynchronous design preserves real-time efficiency even at larger scales. Figure 3(d) analyzes reasoning density by varying the allowed tokens per partial plan; richer plans (more tokens) improve achievement score with only minor impact on throughput.

---

> ### Author Response · Authors · 2025-11-27
> **Response to Reviewer jmVV (3/4)**
>
> **W.2 Relationship to Dreamer and what is novel in Speak-to-Act**
>
> Our controller indeed builds on the Dreamer-V3 RSSM backbone, as we state in Section 4.1. However, we emphasize that Dreamer-V3 is just our controller initialization, not the primary contribution; the main novelty lies in the decoupled VLM planner + language-conditioned world-model controller with asynchronous training and inference protocols.
>
> **Table 1. Ablation on controller architectures.**
> Controller architecture    | Minecraft Diamond
> ---------- | -----
> Dreamer-V3-style RSSM (ours)  | 11.0
> Transformer-based world-model controller [7]     | 11.2
> RNN policy (model-free)     | 9.7
> Value-only world model       | 10.2
>
> While our current paper instantiates the controller as Dreamer-V3, our framework is architecture-agnostic. We do clarify that our framework is architecture-agnostic: in principle, one could plug in a Transformer-based controller or other model-based RL backbones while keeping the same planner interface and training procedure. We have experiments with alternative controllers such as TransDreamer [7] and RNN policies; in the Table above these results suggest similar trends, and we will include these ablations in the revision.
>
> **Q.1 Figure 3: highlighting synchronous vs. asynchronous throughput and performance**
>
> We apologize for the confusion around the terminology. In our framework, online planning corresponds exactly to the asynchronous setting. Offline planning corresponds to the synchronous setting. Figure 3(b) directly compares these two regimes: the online curves use asynchronous planning, while the offline curves use synchronous planning.
>
> **Q.2 Completing Table 1 with stronger SOTA baselines**
>
> **Table 2. Benchmarking with VLA/MARL Baselines.**
> Method                | Atari | Diamond | Crafter | DMLab | Overcooked | Pico Park | MindCraft
> --------                                    | ------| --------| --------| ------| ---------- | ----------| ----------
> Gemma-3-27B            | 880   | 9.7     | 13.8    | 71    | 187.4      | 68.9      | 53.0
> LLaVA-v1.6-34B         | 862   | 9.9     | 12.8    | 74    | 187.2      | 63.5      | 51.6
> Qwen-VL-2.5-72B       | 878   | 11.1    | 13.4    | 77    | 192.3      | 68.4      | 58.5
> GPT-4o           | 891   | 11.7    | 14.1    | 76    | 193.2      | 70.1      | 70.2
> **Ablations**                           |
> Qwen-VL-2.5-72B (Direct VLM Finetune)   | 581   | 10.1    | 7.6    | 45    | 150.1      | 30.6      | 38.2
> GPT-4o (w/o controller)         | 670   | 10.4    | 8.7     | 56    | 180.4      | 50.3      | 50.2
> Controller-Only                             | 809   | 8.2     | 12.6    | 67    | 170.2      | 30.7      | 40.0
> DreamerV3    | 811   | 8.6     | 10.5    | 65    |  -          | -         | -
> MARL Baseline       | -     | -       | -       | -     | 182.5      | 50.8      | 44.9
> **Zero-shot or Promptable**
> MindCraft (Claude)    | -     | -       | -       | -     | -          | -         | 49.0
> Voyage  | -     | 11.8    | -       | -     | -          | -         | -
> RT-2 [1]  (public reproduction)            | 457   | 6.2     | 4.7     | 36    | 124.7  |  34.0  | 33.1
> DEPS [3]               | - | 9.4    |  -  | -  |    -  | -   | 45.6
> LS-Imagine [4]           | -  | 9.6    | -   | -  | -    | -  | 50.1
> JARVIS-1 [6]           | -  | 12.3    | -   | -  | -    | -  | 54.1
> **Finetuned from same Rollout**                           |
> RT-2 [1]  (public reproduction)           | 685   | 9.2     | 8.5     | 58    | 157.5  |  44.2  | 48.3
> DEPS [3]              | - | 10.6    |  -  | -  |    -  | -   | 56.8
> LS-Imagine [4]           | -  | 10.9    | -   | -  | -    | -  | 59.3
> QMIX [5] (public reproduction)              | -   | -    | -    | -  | 187.2      | 58.5      | 53.2
>
> We show more baselines in Table 2 on VLA and MARL baselines. Following the suggestions, we included comparisons with the following VLA and CTDE methods: RT-2, DEPS, LS-Imagine, JARVIS-1, and QMIX. We find that they are competitive in their original domains but our framework outperforms them in general, especially in the more realistic multi-agent collaborative domains, while JARVIS-1 outperforms us in Minecraft Diamond as it is a meticulously designed framework for Minecraft only.
> (Note: We included RT-2 and QMIX’s open reproduction as the baselines. We won’t be able to include results using RT-H as it's not open-source. We can’t run DEPS and LS-Imagine on all tasks as their preset prompts are exclusive for Minecraft.)

---

> ### Author Response · Authors · 2025-11-27
> **Response to Reviewer jmVV (4/4)**
>
> **Q.3 How Table 2 supports “generalization” across instruction modes and planners**
> Section 4.5 studies generalization across two axes: (a) instruction modes (i.e., when instructions are issued to the controller) and (b) generalization to new VLMs given a controller was trained on a particular VLM. Table 2 compares three instruction regimes for each planner (Gemma, Qwen, GPT): fixed cadence, fully VLM-controlled, and VLM-decided proactiveness. All three modes use the same trained controller, and we vary how and when instructions are issued at test time. The fact that performance remains high and changes smoothly across modes (the VLM-decided proactiveness is the best) shows that the controller generalizes across different instruction-timing policies rather than overfitting to a single cadence.
>
> **References**
>
> [1] Zitkovich, Brianna, et al. "Rt-2: Vision-language-action models transfer web knowledge to robotic control." Conference on Robot Learning. PMLR, 2023.
>
> [2] Belkhale, Suneel, et al. "RT-H: Action Hierarchies using Language." Robotics: Science and Systems. 2024.
>
> [3] Wang, Zihao, et al. "Describe, explain, plan and select: interactive planning with llms enables open-world multi-task agents." Advances in Neural Information Processing Systems 36 (2023): 34153-34189.
>
> [4] Li, Jiajian, et al. "Open-World Reinforcement Learning over Long Short-Term Imagination." The Thirteenth International Conference on Learning Representations.
>
> [5] Rashid, Tabish, et al. "Monotonic value function factorisation for deep multi-agent reinforcement learning." Journal of Machine Learning Research 21.178 (2020): 1-51.
>
> [6] Wang, Zihao, et al. "Jarvis-1: Open-world multi-task agents with memory-augmented multimodal language models." IEEE Transactions on Pattern Analysis and Machine Intelligence (2024).
>
> [7] Chen, Chang, et al. "Transdreamer: Reinforcement learning with transformer world models." arXiv preprint arXiv:2202.09481 (2022).
>
> We sincerely thank the reviewers for the feedbacks and will do additional experiments or clarification upon request.

---

### Official Review · Reviewer_6BJx · 2025-11-02

**Soundness:** 2
**Presentation:** 3
**Contribution:** 3
**Rating:** 4
**Confidence:** 4

**Summary:**

Speak-to-Act proposes decoupling embodied decision-making into high-level planning (via VLMs) and low-level control (via trained controllers). The key innovation is an asynchronous inference framework where VLM planners generate language instructions while controllers execute actions in real-time. Controllers are trained using world-model RL (Dreamer-style) with post-hoc language annotation of replay buffer segments. The framework extends naturally to multi-agent settings through language-based coordination, achieving beating baselines on 6/7 tasks.

**Strengths:**

- The decoupling addresses a real problem - VLMs excel at reasoning but struggle with low-latency control, while RL controllers lack semantic understanding. The solution is elegant.
- Beats previous state of the art on 6/7 tasks tested across different environments (Atari, Minecraft, Crafter, etc.), demonstrating broad applicability.
- The online planning mechanism (Algorithm 1) cleverly avoids blocking - VLMs continuously refine plans while controllers execute, reducing latency at instruction boundaries.
- Good analysis of online vs offline planning, model scaling (50M-800M controllers, 2B-72B VLMs), and instruction modes.
-  Parameter-shared controllers with language coordination is simpler than traditional MARL while achieving strong results.

**Weaknesses:**

- The post-hoc VLM annotation is a core contribution, but there's no comparison to simpler alternatives such as random instruction templates or clustering similar action sequences. How much does the VLM annotation quality matter?
- Given the same rollout data, why not directly finetune a VLM to predict actions? This would be the most natural comparison to justify the decomposition.
- Is natural language actually necessary, or would latent goal conditioning work equally well? Compare to VAE/VQ-VAE encodings of trajectory segments (similar to Garg et al. 2022 LISA).
- No training time or compute requirements reported
- When does the approach fail? What happens when VLM instructions are ambiguous or controller misinterprets them? No discussion of error modes.

**Questions:**

- What happens if you replace VLM summarization with: (a) template-based instructions, (b) clustered action patterns with fixed labels, or (c) random instructions? How critical is annotation quality?
- Can you train a VLA baseline using the same data? Specifically, finetune the VLM to directly predict action sequences from observations.
- How does language conditioning compare to latent goal conditioning (e.g., VAE encodings)?
- What are the training times, annotation costs, and inference overheads? How does this scale with environment complexity?
- What types of failures occur?
- Have you tested with human planners providing instructions? This would validate the framework's practical applicability.

---

> ### Author Response · Authors · 2025-11-27
> **Response to Reviewer 6BJx (1/3)**
>
> Thank you for your helpful feedback. We address each of the feedbacks below.
>
> **W.1/Q.1 Impact of VLM post-hoc annotation vs. simpler annotation baselines**
>
> We agree that more explicit ablations would clarify the role of instruction quality. Our current results already hint at robustness to annotation source (Table 3 in the original paper shows similar performance when training with Gemma/Qwen/GPT annotations and evaluating with different planners). Here is our results:
>
> **Table 1. Impact of VLM post-hoc annotation.**
> Train-time instruction type                  | Eval-time planner                | Minecraft Diamond
> --------------------------------------------- | -------------------------------- | -----------------------------------
> Natural language | Qwen-VL-2.5-72B    | 11.1
> Template-based instructions                   | Qwen-VL-2.5-72B      | 9.9
> Clustered action labels (symbolic options)    | Qwen-VL-2.5-72B   | 8.9
> Random strings as "instructions"              | Qwen-VL-2.5-72B   | 8.3
> No instructions (pure RL / world model only)  | No planner | 8.2
>
> One explanation of the performance drop might be the fact that Minecraft Diamond is a long horizon task with a breadth of complexities. Natural language carries more information over its counterparts showing significant advantage.
>
> **W.2/Q.2 Directly fine-tuned VLA baselines from the same rollout data**
>
> **Table 2. Benchmarking with VLA/MARL Baselines.**
> Method                | Atari | Diamond | Crafter | DMLab | Overcooked | Pico Park | MindCraft
> --------                                    | ------| --------| --------| ------| ---------- | ----------| ----------
> Gemma-3-27B            | 880   | 9.7     | 13.8    | 71    | 187.4      | 68.9      | 53.0
> LLaVA-v1.6-34B         | 862   | 9.9     | 12.8    | 74    | 187.2      | 63.5      | 51.6
> Qwen-VL-2.5-72B       | 878   | 11.1    | 13.4    | 77    | 192.3      | 68.4      | 58.5
> GPT-4o           | 891   | 11.7    | 14.1    | 76    | 193.2      | 70.1      | 70.2
> **Ablations**                           |
> Qwen-VL-2.5-72B (Direct VLM Finetune)   | 581   | 10.1    | 7.6    | 45    | 150.1      | 30.6      | 38.2
> GPT-4o (w/o controller)         | 670   | 10.4    | 8.7     | 56    | 180.4      | 50.3      | 50.2
> Controller-Only                             | 809   | 8.2     | 12.6    | 67    | 170.2      | 30.7      | 40.0
> DreamerV3    | 811   | 8.6     | 10.5    | 65    | -          | -         | -
> MARL Baseline       | -     | -       | -       | -     | 182.5      | 50.8      | 44.9
> **Zero-shot or Promptable**
> MindCraft (Claude)    | -     | -       | -       | -     | -          | -         | 49.0
> Voyage  | -     | 11.8    | -       | -     | -          | -         | -
> RT-2 [1]  (public reproduction)            | 457   | 6.2     | 4.7     | 36    | 124.7  |  34.0  | 33.1
> DEPS [3]               | - | 9.4    |  -  | -  |    -  | -   | 45.6
> LS-Imagine [4]           | -  | 9.6    | -   | -  | -    | -  | 50.1
> JARVIS-1 [6]           | -  | 12.3    | -   | -  | -    | -  | 54.1
> **Finetuned from same Rollout**                           |
> RT-2 [1]  (public reproduction)           | 685   | 9.2     | 8.5     | 58    | 157.5 |  44.2  | 48.3
> DEPS [3]              | - | 10.6    |  -  | -  |    -  | -   | 56.8
> LS-Imagine [4]           | -  | 10.9    | -   | -  | -    | -  | 59.3
> QMIX [5] (public reproduction)              | -   | -    | -    | -  | 187.2      | 58.5      | 53.2
>
> We agree that directly fine-tuning a VLM to predict actions is an important baseline. We therefore include **Qwen-VL-2.5-72B (Direct VLM Finetune)** in Table 2; the former use the same rollout data but act directly from observations to actions. Across environments, these direct-policy VLMs perform substantially worse than our decomposed planner+controller system. We believe this reflects a fundamental mismatch: large VLMs are optimized for language reasoning, not 10–60 Hz control with environment-specific action spaces, which leads to latency and sample-efficiency issues, especially in multi-agent settings.
> Moreover, we included finetuned VLA and MARL CTDE frameworks at the bottom part of the table for more baselines.

---

> ### Author Response · Authors · 2025-11-27
> **Response to Reviewer 6BJx (2/3)**
>
> **W.3/Q.3 Is natural language necessary? Comparison to latent goal conditioning (e.g., LISA)**
>
> We appreciate this suggestion and will cite Garg et al. (2022, LISA) as a relevant alternative. We will mak a comparison with it on latency and performance in the next version of the paper due to time constraints. Our choice of natural language is motivated by (1) Language lets both humans and different VLMs participate as planners without retraining. (2) Our core motivation stems from the fact that our framework is plug-in-and-play with easy use of any combos of VLMs. Requiring latent goal conditioning would not allow us to evaluate the zero-shot planning or communication abilities of VLMs.
>
> **W.4/Q.4 Training compute, annotation cost, and inference overhead**
>
> Our experiments were run on 4× NVIDIA RTX A6000 GPUs and an Intel Xeon Gold 6338 CPU @ 2.00GHz. Across all tasks, the maximum training time for a single environment is ~36 hours wall-clock, and the average per-task training time is ~23 hours. In the revision, we will add an appendix table summarizing GPU hours and wall-clock time per environment. The annotation does not significantly bring additional overhead on top of standard RL replay collection as it is done while training. The annotation overhead brings 17% additional GPU hours and 12% speed decrease in training, which is comparably small. Inference overhead is described in the Figure 3 with each environment step inference comparable to a light-weight RNN inference. Note that the computation overhead does not scale with environment complexity. By complexity, environment observations or task difficulty do not increase inference speed and only the performance of the model. Complex observations only bring in additional low-dimentional inputs and task complexity only requires agents for longer-horizon reasoning which is dependent on VLM's reasoning.
>
> **W.5/Q.5 Failure modes under ambiguous or incorrect instructions**
>
> We appreciate the request for a more explicit error-mode discussion and agree this was under-emphasized. We will upload a new version of paper to focus on this aspect as well.
> For the three of the domains, we sampled 200 random instructions issued by a planner, and manually evaluated the instruction-following accuracy of that domain’s controller (i.e., whether the controller executed the instruction exactly as it was given). On Minecraft Diamond for example, the instruction-following accuracy of the controller is around 96% across different planners using GPT-4o in instruction annotator. This suggests that the controller reliably follows the planner most of the time.
>
> **Table 3. Instruction Success Rate.**
> Tasks | Gemma-3-27B | Qwen-VL-2.5-72B | GPT-4o
> ------ | -------------- | -------| -------
> Minecraft Diamond | 193/200 | 190/200 | 194/200
> Pico Park        | 186/200 |  180/200 | 189/200
> Atari            | 181/200 | 172/200 | 185/200
>
> Table 3 in the original paper shows cross-planner generalization: training instructions with one planner (Gemma, Qwen, GPT) and evaluating with another yields similar achievement scores. This shows that our approach results in controllers that do not favor particular VLMs as planners and verifies the plug-and-play nature of our framework.
> Only half of the replay buffer is annotated with instructions; the remaining half is pure RL without any language labels. This design explicitly limits the impact of noisy or biased instructions. At inference time, if the planner issues a suboptimal instruction, the controller may still deviate when the environment reward structure makes that instruction consistently low-value; we observe such “self-correction” especially in repeated reward-dense settings (e.g., Atari games with strong local feedback).
> The most common controller failure we observe is how it handles ambiguous or very generic instructions (e.g., “explore a bit more”, “try something different”). This can lead to locally suboptimal behaviors until a clearer instruction arrives. However, because the planner runs asynchronously, it can quickly issue a refined instruction once it observes that progress stalls.
>
> **Q.6 Human planners as instruction agents**
>
> We agree that human planners are a natural and practically important use case. Our system is designed precisely so that the planner π_p is a black-box function from observations (and optional history) to natural-language instructions; replacing the VLM with a human fits seamlessly into the same interface. Setting up a human study is beyond the scope of our paper right now. As noted earlier, we manually looked at the instruction-following ability of the controller which shows our framework robust human instructability. However, the prompt usually prompts VLMs with a distribution of instruction length for 6–18 tokens, imperative, verifiable from observations. This means the instructions are explicitly chosen to align with how a human might naturally instruct a teammate.

---

> ### Author Response · Authors · 2025-11-27
> **Response to Reviewer 6BJx (3/3)**
>
> **References**
>
> [1] Zitkovich, Brianna, et al. "Rt-2: Vision-language-action models transfer web knowledge to robotic control." Conference on Robot Learning. PMLR, 2023.
>
> [2] Belkhale, Suneel, et al. "RT-H: Action Hierarchies using Language." Robotics: Science and Systems. 2024.
>
> [3] Wang, Zihao, et al. "Describe, explain, plan and select: interactive planning with llms enables open-world multi-task agents." Advances in Neural Information Processing Systems 36 (2023): 34153-34189.
>
> [4] Li, Jiajian, et al. "Open-World Reinforcement Learning over Long Short-Term Imagination." The Thirteenth International Conference on Learning Representations.
>
> [5] Rashid, Tabish, et al. "Monotonic value function factorisation for deep multi-agent reinforcement learning." Journal of Machine Learning Research 21.178 (2020): 1-51.
>
> [6] Wang, Zihao, et al. "Jarvis-1: Open-world multi-task agents with memory-augmented multimodal language models." IEEE Transactions on Pattern Analysis and Machine Intelligence (2024).
>
> We sincerely thank the reviewers for the feedbacks and will do additional experiments or clarification upon request.

---

### Author Response · Authors · 2025-12-03
**Response to AC and Reviewers**

Dear AC and reviewers,

We would like to briefly summarize how we have addressed all reviewer questions and concerns during the rebuttal.
Here is a summary of what we added and clarified in the rebuttal:

**Core contribution & architecture.** We clarified that the main contribution is a plug-and-play framework that
  - trains language-conditioned world-model controllers via post-hoc VLM annotation of replay segment
  - allows arbitrary pretrained VLM planners (without any finetuning) to be swapped in for single- or multi-agent embodied tasks. Dreamer-V3 is only used as a controller backbone; the framework is controller-architecture-agnostic.

**Expanded baselines on VLA & MARL.** We added a comprehensive comparison table including:
  - VLA: RT-2, DEPS, LS-Imagine, JARVIS-1, plus direct VLM finetuning baselines ("Direct VLM Finetune", "GPT-4o w/o controller").
  - MARL/CTDE: QMIX and a DreamerV3 controller-only baseline.

Using the same rollout data, the decomposed planner+controller setup generally outperforms these methods across most tasks, with JARVIS-1 competitive only in Minecraft.

**Language annotation ablations.** We added explicit ablations on instruction type on natural language vs template-based vs clustered option labels vs random strings vs no instructions. Natural language gives the best performance, but controllers still outperform pure RL even with weaker instruction sources, indicating robustness to annotation quality.

**Controller architecture ablations.** We added new experiments replacing the Dreamer-style RSSM with:
  - a Transformer-based world-model controller,
  - an RNN model-free policy, and
  - a value-only world model.

Results show similar trends for both RSSM and Transformer controllers, with weaker performance for the RNN and value-only variants, supporting that the framework is not tied to a specific controller design.

**Instruction timing & generalization.** We tested on multiple instruction modes using the same trained controller:
  - fixed cadence,
  - fully planner-controlled cadence,
  - planner-decided proactiveness.

Overall, the performance remains high and changes smoothly across modes, showing the controller generalizes across different instruction-timing policies instead of overfitting to a single schedule.

**Efficiency & online vs offline planning.** We reported:
  -Training resources (4× A6000, wall-clock hours per environment) and that annotation adds only modest overhead (~17% GPU hours, ~12% slowdown).
  - Inference throughput comparisons vs VLA baselines, showing higher throughput because only the small controller runs at control frequency, while the VLM planner is invoked sparsely.
  - A direct online (same as asynchronous) vs offline (same as synchronous) comparison referenced at original paper Fig.3: asynchronous planning yields higher throughput and modest performance gains, while scaling controller (50M to 800M) and VLM (2B to 72B) size improves returns with near constant throughput.

**Failure modes, reward–instruction interaction.**
We show results on instruction-following accuracy (with ~90–96% across domains/planners) and described typical failure cases especially under very generic instructions. We also clarified that:
  - environment reward remains the sole optimization target (no language-based reward),
  - in dense-reward settings, RL gradients override bad instructions,
  - in sparse-reward settings, Dreamer-style value learning plus auxiliary instruction labels still drives learning.

- Multi-agent scaling. We provided:
  - comparison of centralized vs decentralized communication modes (higher success and similar throughput in decentralized setups as described in the original paper Fig.3)
  - discussion of multi-agent interaction patterns including role negotiation, bottlenecks
  - clarified that current strong scalability are just for symmetric agents with shared controllers. Also, adding role-conditioned controllers and planner prompts are a simply straightforward follow-up work to asymmetric settings.

Overall, we are happy to address any follow-up concerns upon request and hopefully the above can address reviewers' concerns and questions.

---

### Meta-Review · Area_Chair_x5kc · 2026-01-06

**Summary:**

The reviewers acknowledge the reasonable and elegant design, including its asynchrony and modularity, the strong performance, as well as the extensive benchmarking and evaluations.

However, consistent concerns center on its limited conceptual novelty and unclear positioning, insufficient comparisons and analysis with more baselines and in more settings, and incomplete analysis of computational costs, failure modes, and scalability limits.

**Reviewer Concerns:**

- The rebuttal addressed several issues, including experiments with more baselines (e.g., VLAs, a fine-tuned VLM, etc.) and ablation studies (e.g., controllers, annotations, zero-shot, etc.), as well as different settings of instruction modes and failure modes.
- The rebuttal also clarified training cost, throughput advantages, and robustness to instruction noise.


- However, it emphasized engineering integration and plug-and-play flexibility without convincingly distinguishing conceptual novelty from prior hierarchical VLA/MARL frameworks, which is a major remaining concern.

- Compute overhead in multi-agent VLM use and the per-agent VLM inference cost are not unquantified, which may be the drawback and undermine practical scalability claims in realistic MARL deployments.

- Unlike some compared baselines, the multi-task setting remains unaddressed for the proposed method.

Given these unresolved issues, particularly on novelty and scalability, the paper remains borderline and leans toward rejection.

**Reviewer Scores:**

- 6BJx may keep or increase the score from 4 to 6. The questions related to additional experiments are mostly addressed; however, inference overhead, scalability, and human planners are not.

- jmVV may keep or increase the score from 4 to 6, as the rebuttal provided several additional experiments. Novelty discussion and in-depth analysis were included but may not be sufficiently clear.

- eR51 is unlikely to change the score, as the main concerns about the novelty and multi-task generalization are not fully addressed.

- Kt4V is unlikely to change the score, as conceptual novelty and the missing analysis of VLM inference overhead are still not fully addressed.

---

### Decision · Program_Chairs · 2026-01-26

Reject